# Factors associated with soil-transmitted helminths infection in Benin: Findings from the DeWorm3 study

**Euripide F. G. A. Avokpaho**[1,2]*, **Parfait Houngbégnon**[1], **Manfred Accrombessi**[1,3], **Eloïc Atindégla**[1], **Elodie Yard**[4], **Arianna Rubin Means**[5,6], **David S. Kennedy**[4,3], **D. Timothy J. Littlewood**[4], **André Garcia**[7], **Achille Massougbodji**[1], **Sean R. Galagan**[5,6], **Judd L. Walson**[5,6], **Gilles Cottrell**[7], **Moudachirou Ibikounlé**[1,8‡], **Kristjana Hrönn Ásbjörnsdóttir**[5,9,10‡], **Adrian J. F. Luty**[7‡]

1 Institut de Recherche Clinique du Bénin, Abomey-Calavi, Benin, 2 Université de Paris, ED 393 Pierre Louis de Santé Publique, Paris, France, 3 Faculty of Infectious and Tropical Diseases, London School of Hygiene & Tropical Medicine, London, United Kingdom, 4 DeWorm3, Department of Life Sciences, Natural History Museum, London, United Kingdom, 5 DeWorm3, University of Washington, Seattle, Washington, United States of America, 6 Department of Global Health, University of Washington, Seattle, Washington, United States of America, 7 Université de Paris, MERIT, IRD, Paris, France, 8 Centre de Recherche pour la lutte contre les Maladies Infectieuses Tropicales (CReMIT/TIDRC), Université d'Abomey-Calavi, Abomey-Calavi, Bénin, 9 Department of Epidemiology, University of Washington, Seattle, Washington, United States of America, 10 Centre for Public Health Sciences, University of Iceland, Reykjavík, Iceland

‡ These authors are joint co-last authors on this work.
* euripideavokpaho@gmail.com

**Editor:** jong-Yil Chai, Seoul National University College of Medicine, REPUBLIC OF KOREA

**Data Availability Statement:** Under agreement with the IRBs of the study, data must be blinded until the study concludes. Therefore, to avoid

## Abstract

### Background

Despite several years of school-based MDA implementation, STH infections remain an important public health problem in Benin, with a country-wide prevalence of 20% in 2015. The DeWorm3 study is designed to assess the feasibility of using community-based MDA with albendazole to interrupt the transmission of STH, through a series of cluster-random-ized trials in Benin, India and Malawi. We used the pre-treatment baseline survey data to describe and analyze the factors associated with STH infection in Comé, the study site of the DeWorm3 project in Benin. These data will improve understanding of the challenges that need to be addressed in order to eliminate STH as a public health problem in Benin.

### Methods

Between March and April 2018, the prevalence of STH (hookworm spp., *Ascaris* and *Trichuris trichiura*) was assessed by Kato-Katz in stool samples collected from 6,153 residents in the community of Comé, Benin using a stratified random sampling procedure. A standard-ized survey questionnaire was used to collect information from individual households concerning factors potentially associated with the presence and intensity of STH infections in pre-school (PSAC, aged 1–4), school-aged children (SAC, aged 5–14) and adults (aged 15 and above). Multilevel mixed-effects models were used to assess associations between these factors and STH infection.

breaching the agreement with the ethical approval bodies, data cannot be shared publicly because the study remains blinded to outcome data. Data are available from the DeWorm3 Institutional Data Access Committee (contact via dw3data@uw.edu) for researchers who meet the criteria for access to these data.

**Funding:** JLW and DTL received the DeWorm3 study funding from The Bill and Melinda Gates foundation (OPP1129535). https://www.gatesfoundation.org/. EFGAA is a PhD candidate at the University of Paris. His research is funded by DeWorm3 as a staff member of the Benin coordinating team, and by the French Research Institute for Sustainable Development (IRD) through the International Mixed Laboratory LMI CONS_HELM (helminth infections: treatments and consequences on health and development in the South). https://www.ird.fr/benin/partenariat. The funders had no role in study design, data collection and analysis, decision to publish, or preparation of the manuscript.

**Competing interests:** The authors have declared that no competing interests exist.

## Results

The overall prevalence of STH infection was 5.3%; 3.2% hookworm spp., 2.1% *Ascaris lumbricoides* and 0.1% *Trichuris*. Hookworm spp. were more prevalent in adults than in SAC (4.4% *versus* 2.0%, respectively; p = 0.0001) and PSAC (4.4% *versus* 1.0%, respectively; p<0.0001), whilst *Ascaris lumbricoides* was more prevalent in SAC than in adults (3.0% *versus* 1.7%, respectively; p = 0.004). Being PSAC (adjusted Odds Ratio (aOR) = 0.2, p< 0.001; adjusted Infection Intensity Ratio (aIIR) = 0.1, p<0.001) or SAC (aOR = 0.5, p = 0.008; aIIR = 0.3, p = 0.01), being a female (aOR = 0.6, p = 0.004; aIIR = 0.3, p = 0.001), and having received deworming treatment the previous year (aOR = 0.4, p< 0.002; aIIR = 0.2, p<0.001) were associated with a lower prevalence and intensity of hookworm infection. Lower income (lowest quintile: aOR = 5.0, p<0.001, $2^{nd}$ quintile aOR = 3.6, p = 0.001 and $3^{rd}$ quintile aOR = 2.5, p = 0.02), being a farmer (aOR = 1.8, p = 0.02), medium population density (aOR = 2.6, p = 0.01), and open defecation (aOR = 0.5, p = 0.04) were associated with a higher prevalence of hookworm infection. Lower education—no education, primary or secondary school- (aIIR = 40.1, p = 0.01; aIIR = 30.9, p = 0.02; aIIR = 19.3, p = 0.04, respectively), farming (aIIR = 3.9, p = 0.002), natural flooring (aIIR = 0.2, p = 0.06), peri-urban settings (aIIR = 6.2, 95%CI 1.82–20.90, p = 0.003), and unimproved water source more than 30 minutes from the household (aIIR = 13.5, p = 0.02) were associated with a higher intensity of hookworm infection. Improved and unshared toilet was associated with lower intensity of hookworm infections (aIIR = 0.2, p = 0.01). SAC had a higher odds of *Ascaris lumbricoides* infection than adults (aOR = 2.0, p = 0.01) and females had a lower odds of infection (aOR = 0.5, p = 0.02).

## Conclusion

Hookworm spp. are the most prevalent STH in Comé, with a persistent reservoir in adults that is not addressed by current control measures based on school MDA. Expanding MDA to target adults and PSAC is necessary to substantially impact population prevalence, particularly for hookworm.

## Trial registration

ClinicalTrials.gov NCT03014167.

### Author summary

Despite several years of deworming campaigns targeting school-aged children, soil-transmitted helminths (STH) remains a public health problem in most developing countries, including Benin. The burden is mostly on children and pregnant women, but also on the whole society. Soil-transmitted helminths are responsible for malnutrition, anemia, low birth weight, cognitive impairment, decrease of school performance, and subsequently economic loss. The current strategy of the Benin National Neglected Tropical Diseases (NTD) Program is to achieve STH control through mass drug administration campaigns targeting school-aged children (SAC). The baseline data of Deworm3 study, implemented in Comé, southern Benin, as part of a multicountry (Benin, Malawi and India) STH elimination trial, shows that previous school deworming campaigns decreased STH prevalence;

however there is a persistent reservoir of STH infection in adults and pre-school aged children that should be targeted for a better impact. In order to eliminate STH as a public health problem, Benin National NTD Program would need to increase its target population, from the SAC to the whole community. The future results of Deworm3 trial would demonstrate whether the STH elimination goal STH using community wide mass drug administration would be achievable.

## Introduction

Soil-transmitted helminths (STHs) infections are among the most common infections worldwide, affecting more than 1.5 billion of the poorest and most marginalized communities globally. [1] The most common STHs infecting humans are *Ascaris lumbricoides*, *Trichuris trichiura* and the hookworm species, *Necator americanus* and *Ancylostoma duodenale*. Soil-transmitted helminths are transmitted by eggs present in human feces which in turn contaminate soil and water in areas with poor sanitation, conditions often found in low-resource countries. [2] Soil-transmitted helminths are widely distributed in tropical and subtropical areas, with the greatest numbers occurring in sub-Saharan Africa, the Americas, China and South-East Asia. [3]

The World Health Organization (WHO) considers STHs a public health problem in areas where >1% of the at-risk population has moderate-to-heavy intensity infections, measured by the number of eggs per gram of stool counted during the stool examination [4]. These moderate to high intensity helminth infections are associated with poor cognitive and motor outcomes in infants, as well as with anemia. [5–9] Pre-school children (PSAC), school age children (SAC) and women of reproductive age (WRA), including adolescent girls, pregnant women, lactating women, and non-pregnant and non-lactating women living in endemic areas, are at highest risk of morbidity due to STHs. Clear policy and guidance are essential to support country-level efforts to expand routine deworming of WRA, and recent WHO publications have provided the necessary policy framework. [2,3]

The WHO Neglected Tropical Disease (NTD) Roadmap and London Declaration have accelerated the progress toward eliminating selected NTDs, such as lymphatic filariasis and onchocerciasis, and formalized long-term disease-specific goals for other NTDs. [10] Global interest is shifting from control towards an elimination strategy for other NTDs, including the possibility of breaking the transmission of STHs through community-wide mass drug administration (MDA). [11]

In Benin, all major STHs are a recognized public health problem, with more than 50% of districts requiring MDA based on the results of a recent national mapping done from 2013 to 2015 that sampled stool from SAC. [12,13] In the Comé District, the 2013–2015's national mapping showed a prevalence of STHs in SAC of 20%, despite three rounds of yearly school-based MDA with albendazole in 2015 (coverage 59%), 2016 (coverage 78%) and 2017 (coverage 83%). [13]

In 2017, the DeWorm3 project (ClinicalTrials.gov Identifier NCT03014167) was initiated in Benin, and, in parallel, in India and Malawi. Using a cluster randomized controlled study design, the primary objective of the project is to determine whether the provision of an enhanced (twice yearly) level of high-coverage MDA, targeting all age groups in a whole community over a 3-year period, can interrupt transmission of STHs [11]. Here we report analyses of baseline data from a longitudinal monitoring cohort randomly selected from the whole population involved in the trial in order to determine the demographic and other parameters potentially associated with the STHs infections detected by microscopy using a standard Kato-Katz procedure.

## Materials and methods

### Ethics statement

Ethical approval of the DeWorm3 trial protocol was obtained both from the Human Subjects Division at the University of Washington and the National Ethics Committee for Health Research of Benin (CNERS ethical clearance reference No: 002-2017/MS/DC/SGM/DFR/CNERS-Ministry of Health, Benin). The trial was registered at Clinical Trials.gov NCT03014167 (https://clinicaltrials.gov/ct2/show/NCT03014167) [14]. Written consent was obtained from each participant (or participants' parents, when participants were under 18 years of age). For children aged 1–6 years old, verbal assent was obtained and for adolescents aged 7–17 years written assent was obtained. Data were collected electronically using password protected smartphones and was stored in datasets. Although WHO guidelines do not recommend MDA for adults, following the stool analysis any adults ($\geq$15 years of age) in control clusters presenting moderate-to-heavy intensity STH infection according to WHO definitions [15] or requiring treatment according to local guidelines, were treated with albendazole by study staff.

### Study area and population

The DeWorm3 trial in Benin is being conducted in the district of Comé. (Fig 1) The study site selection was based on criteria that have previously been reported. [16] Comé is located 70 km west of Cotonou in the Mono department, at latitude 6˚24′N and longitude 1˚53′E. The district covers an area of 153 km$^2$ with a population estimated at 79,989 inhabitants in the census of 2012, and an estimated yearly growth rate of 2.1%. [17] The district has five sub-districts (Central Comé, Akodéha, Oumako, Agatogbo and Ouèdèmè-Pedah) subdivided into 52 villages/areas or neighborhoods. The climate is sub-equatorial, tropical, alternating between two rainy seasons (April to July and September to November) and two dry seasons (December to March and August). Rainfall varies between 900 and 1,200 mm per year. The population of Comé is mostly rural, with agriculture and fishing as the main activities. The agricultural, livestock breeding, market gardening and fish farming areas cover 73% of the territory. The commune of Comé is crossed by a dense hydrographic network with 22% of floodable zones.[18].

### Study design

The protocol and aims of the DeWorm3 study have been published elsewhere. [11] A baseline census was conducted from January 8th to February 9th, 2018 followed by cluster demarcation. The geospatial locations of all households were mapped using ArcGIS (Redlands, CA), and the study area was divided into 40 clusters with between 1,650 and 4,000 residents per cluster. From March 6th to April 5th, 2018, 6000 individuals (150 individuals by cluster) were randomly selected to constitute a longitudinal monitoring cohort (LMC) participating in annual follow-up STHs infection surveys over 5 years. The LMC was selected from the censused population using stratified random sampling of PSAC aged 1–4 years old, SAC aged 5–14 years old and participants aged 15 years old and above (considered as adults in this study), at a ratio of 1:1:3. A sampling list of 150 individuals (i.e. 30 PSAC, 30 SAC and 90 adults) was initially generated and backup lists of 75 individuals were issued to replace participants who could not be located or refused to participate. We are reporting in this paper the findings of the baseline cross-sectional survey, the first of the five annual surveys that will we done as part of the LMC, using the STROBE checklist [19] (S1 STROBE checklist). Participants were interviewed about individual-level STHs risk factors, including a survey of self-reported WASH access and use, history of deworming, and direct observation of WASH facilities and participants' use of

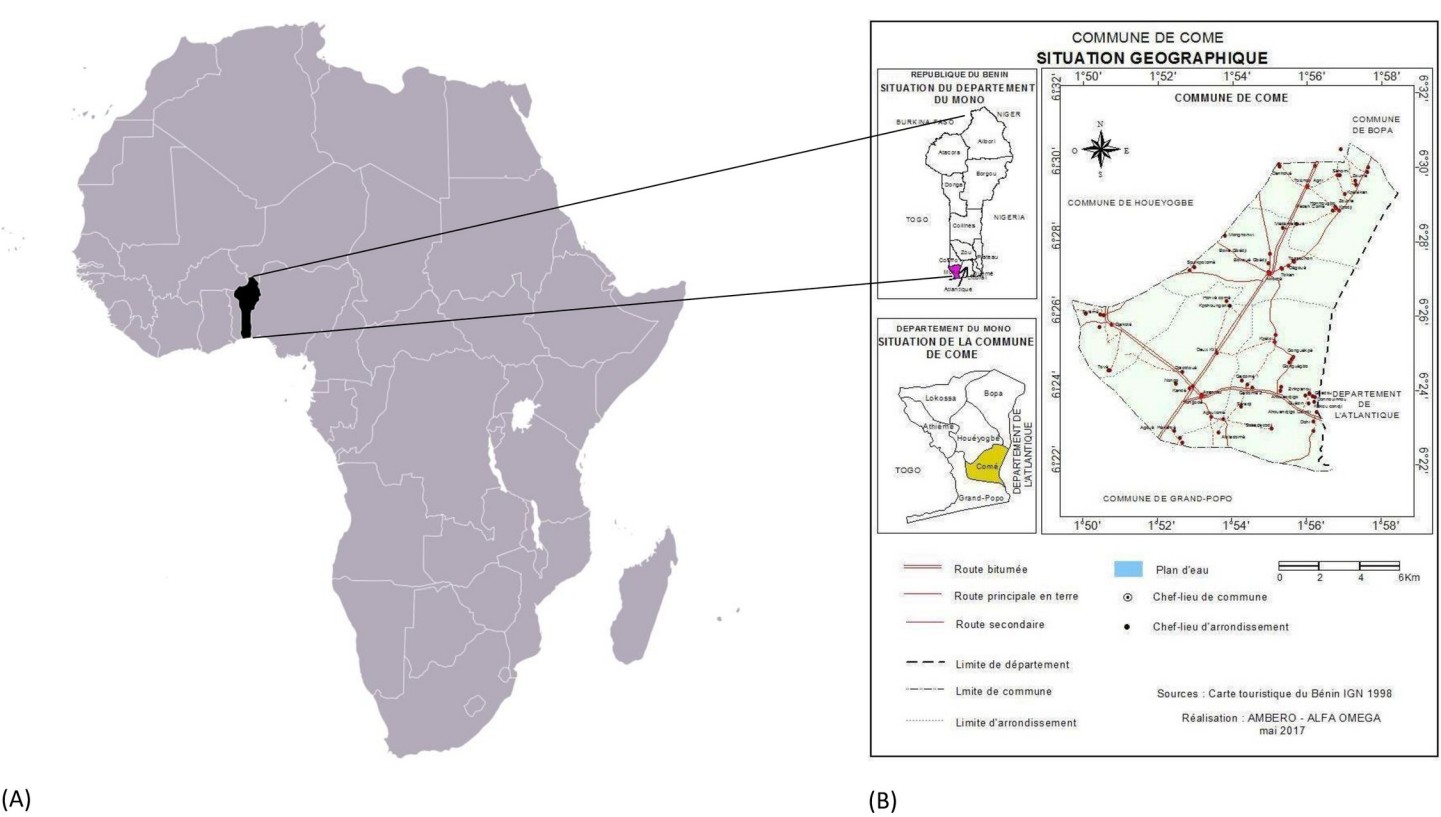

**Fig 1. Geographical location of Comé district, in Benin republic, West Africa.** (A) Location of Benin republic in Africa (https://www.mapsland.com/africa/benin/large-location-map-of-benin-in-africa). (B) Administrative boundaries of Comé district. Adapted by AMBERO-ALPHA OMEGA, May 2017. From Benin Tourism Card, National Geographic Institute 1998.[18].

footwear. Individuals participating in the LMC agreed to provide one stool sample for immediate analysis using the Kato-Katz method [20].

## Data collection

**Kato-Katz data.** Stool samples from LMC participants were collected by study staff and delivered to the laboratory within one hour. Samples were screened using the Kato-Katz technique. Two slides were prepared from each sample, and each slide was examined by two experienced lab technicians who recorded egg counts for each parasite separately. Prevalence was calculated both for individual STHs types and cumulatively according to the following formulas:

- - The prevalence per STH type:

$$p = \frac{\text{Number of samples where at least one egg of STH species is found}}{\text{total number of samples examined}} \text{ x } 100$$

- - The cumulative STH prevalence:

$$p = \frac{\text{Number of positive samples for one, two or three STH species}}{\text{total number of samples examined}} \text{ x } 100$$

In cases of co-infection, prevalence and intensity were assessed separately for each species.

The sample was counted to calculate the cumulative prevalence. The parasite intensity was calculated from a Kato-Katz smear made with 41.7 mg of stool, by multiplying the egg count from the slide by a factor of 24 (24 x 41.7 mg $\approx$ 1 g) to get the number of eggs per gram of stool (EPG).

After stool samples collected from enrolled LMC participants were tested by Kato-Katz (KK), they were aliquoted into three samples for storage and eventual quantitative polymerase chain reaction (qPCR) testing. Quality assurance (QA) was conducted to ensure data quality of KK testing. A subset of 10% of slides was randomly selected for reading by the laboratory supervisor and compared against the original readings (S1 Table). Slides were chosen randomly for checking via the SurveyCTO software. All slides reported as positive were cross-checked by the supervisor to confirm the STH species reported by each reader. Routine checks of a selection of slides reported as negative were also carried out by the supervisor for verification.

## Outcomes

The primary outcomes were individual-level infection status for each STH type (positive / negative) and intensity of infection.

## Variables

Individual factors (including age, gender, history of deworming during the past year and shoe wearing behavior), household factors (including highest educational level achieved, head of household occupation, household asset index, urbanization), water sanitation and hygiene (WASH) factors (household water service, household sanitation, household hand washing facility) and environmental factors (elevation, soil sand fraction, soil acidity at average depth [0-5-15 cm], MODIS [Moderate Resolution Imaging Spectroradiometer] daytime land surface temperature mean for 2018 [°Celsius], MODIS Enhanced Vegetation Index [EVI] mean for 2018, MODIS normalized difference vegetation index [NDVI] mean for 2018, aridity index) were collected or constructed based on existing data.

**Water, sanitation and hygiene variables.** Water sources and sanitation facilities reported were grouped and categorized according to the 2017 WHO/UNICEF Joint Monitoring Program (JMP) methodology (none, improved, unimproved, limited or basic). [21] Improved drinking water sources are those that have the potential to deliver safe water by nature of their design and construction, while improved sanitation facilities are those designed to hygienically separate excreta from human contact. [22] Distance from the household to the closest water source and sharing status for sanitation were also collected.

**Asset index.** An asset index was compiled using principal components analysis. The procedure described by the Demographics and Health Survey *(Steps to constructing the new DHS Wealth Index)*[23] was followed, but factors associated with STHs transmission (crowding [residents/room], WASH variables included in the risk factors analysis, and flooring materials) were excluded as they were evaluated separately in the model.

**Environmental variables.** We examined the association of the following environmental and sociodemographic factors with STH infection: mean enhanced vegetation index and land surface temperature during the study period; elevation; aridity; soil acidity and sand content; and population density. These environmental, topographical, and sociodemographic measures were extracted for each household using point-based extraction using ArcGIS 10.3 (Environmental Systems Research Institute Inc., Redlands, CA, USA). Data sources and methods have been described previously [24]. Estimates of population density were obtained by calculating the number of individuals living within 1km$^2$ buffer around each household, which was used to classify areas as high, medium or low population density. Continuous variables were categorized by tertiles for analysis.

## Descriptive statistics

Categorical variables were described using proportions and 95% confidence intervals and the continuous variables were described by the median and interquartile ranges. To compare proportions, we used the Chi-square, and Cuzick trend tests. Continuous variables were compared using the Student T-test and analysis of variance (ANOVA).

For each STH species we determined the cluster level prevalence (proportion of individuals infected in the cluster) and cluster level arithmetic mean of individual's egg density per gram of feces. We plotted the cluster level mean egg density against the cluster level prevalence and assessed the strength of the linear relationship using Pearson's correlation coefficient test. Descriptive statistics were generated using Stata 14.0 (Stata Corp, College Station, Texas).

## Factors associated with STH infection

Factors associated with presence and intensity of baseline infection with each STH species were identified using mixed effects models with random effects at the household and cluster levels and exchangeable correlation matrix. For binary infection status, mixed effects logistic regression was used, while for intensity of infection negative binomial mixed effects regression was used.

For the negative binomial regression, the output was the infection intensity ratio (IIR):

$$\text{IIR} = e^{\beta} = e^{[\log(\mu_{x0+1}) - \log(\mu_{x0})]} = e^{[\log(\mu_{x0+1}/\mu_{x0})]}$$

where $\beta$ is the regression coefficient, $\mu$ is the expected intensity of infection (EPG) and the subscripts represent where the predictor variable, say x, is evaluated at $x_0$ and $x_{0+1}$ (implying a one unit change in the predictor variable x). The IIR are interpreted as the ratio of expected intensity of infection for a one unit increase in the predictor variable given the other variables are held constant in the model.

All models were adjusted for age and sex. Groups of socio-economic status indicators, environmental factors and WASH factors hypothesized to be associated with infection were proposed *a priori* in the multivariable analysis. For groups of indicators with similar variables, the factor from each group with the lowest Akaike Information Criterion (AIC) in univariate analyses was selected for inclusion in the multivariable model. Models were further simplified by backward stepwise elimination until AIC was no longer further reduced in the adjusted model. Random effects predicted by the fully adjusted model were compared to those predicted by a model containing only age and sex and the proportion of clustering explained by the explanatory variables was quantified.

## Results

### Descriptive

Based on the census data, 11,979 individuals were selected for participation in three consecutive stages (Stage 1: n = 5,979; Stage 2: n = 3,000; Stage 3: n = 3,000), with the goal to reach 150 individuals in each cluster: 30 PSAC, 30 SAC and 90 adults. Characteristics of the LMC population in comparison to censused population of the DeWorm3 site are presented in Table 1. Fig 2 presents the study flow chart. Individuals selected were listed as living in 9,265 households from which 8,741 were located and visited. In those households 7,045 individuals were present, among whom 6,814 consented to participate in the LMC cohort. Stool samples were collected from 6,153 individuals. The most common reasons for stool samples not being collected were (i) no sample visit documented (319), (ii) sample could not be collected after three visits (n = 111), (ii) refusal to provide sample (n = 231). As no documented survey could be verified for 14 individuals, Kato-Katz tests performed were confirmed for 6,139 samples

**Table 1. Comparison of censused population of the DeWorm3 site and longitudinal monitoring cohort (LMC).**

|  | Census | LMC |
|---|---|---|
|  | n (%)** / median (IQR)*** | n (%)** / median (IQR)*** |
| **Study population** | 94,969 | 6,139 |
| Gender* |  |  |
| - Female | 49,081 (51.7) | 3,311 (54.0) |
| - Male | 45,888 (48.3) | 2,828 (46.1) |
| Age distribution* |  |  |
| - Infants (<1 years) | 2,616 (2.7) | - |
| - Preschool-age children (1–4 years) | 11,188 (11.8) | 1,184 (19.3) |
| - School age children (5–14 years) | 26,043 (27.4) | 1,335 (21.8) |
| - Adults (15+ years) | 54,882 (57.8) | 3,620 (59.0) |
| **Household characteristics** |  |  |
| Roof materials* |  |  |
| - Natural materials | 5,311 (5.6) | 349 (5.7) |
| - Man-made materials | 89,342 (94.1) | 5,771 (94.0) |
| Walls materials |  |  |
| - Natural materials | 22,200 (23.4) | 1,359 (22.1) |
| - Man-made materials | 71,258 (75.0) | 4,665 (76.0) |
| Flooring materials |  |  |
| - Natural materials | 16,336 (17.2) | 950 (15.5) |
| - Man-made materials | 78,200 (82.3) | 5,162 (84.1) |
| Sources of income<br>-categories |  |  |
| Asset Index quintiles **** | (n = 24,378 households) | (n = 6,139 individuals) |
| Quintile 1: range [-2.7;-1.8[ | 5,243 (21.5) | 985 (16.0) |
| Quintile 2: range [-1.8;-1.2[ | 4,620 (18.9) | 1,043 (17.0) |
| Quintile 3: range [-1.2;-0.2[ | 4,840 (19.8) | 1,175 (19.1) |
| Quintile 4: range [-0.2; 2.0[ | 4,884 (20.0) | 1,378 (22.4) |
| Quintile 5: range [2.0; 12.0] | 4,791 (19.6) | 1,558 (25.4) |
| Number of Residents per Household | 5 (4–7) | 5 (4–7) |

Notes

*Missing <5% unless otherwise specified.

**for categorical variables

***for continuous variables

**** These numbers represent ranges of the quintiles of the asset index, a wealth score constructed using Demographic and Health Survey methods. The lowest quintiles represent poor wealth asset (poorest) and the highest represent good wealth asset (richest).

comprising 1,184 PSAC (98.7% of 1,200 expected), 1,335 SAC (>100% of 1,200 expected), and 3,620 adults (>100% of 3,600 expected). In total 6,139 tests had two slides read by lab technicians. A random subset of Kato-Katz tests was selected for reading by the supervisor and compared against the original readings for quality assurance. Agreement between original Kato-Katz reading and QA reading was 99.3%. (S1 Table).

## Prevalence of STHs

Among the 6,139 individuals tested by Kato-Katz, STHs infections of any type were found in 324 (5.3%), 199 (3.2%) due to hookworm and 126 (2.0%) due to *Ascaris lumbricoides*. *Trichuris trichiura* was only detected in five (0.1%) individuals (Table 2). Six individuals were co-

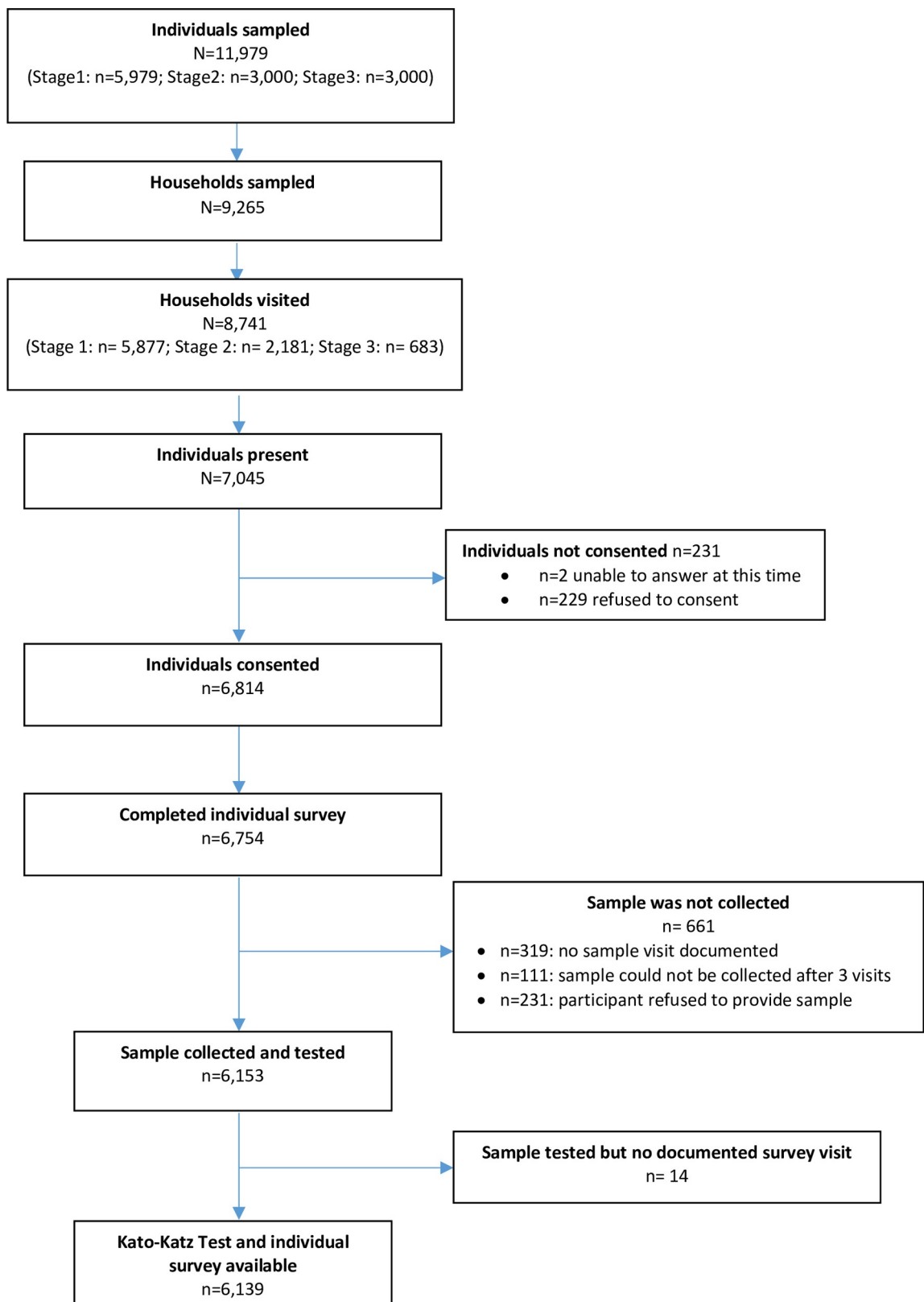

**Fig 2. Flow diagram of stool sample collection for Benin site DeWorm3 baseline prevalence survey in Comé.**

**Table 2. Unweighted STH prevalence and intensity of infection by Kato-Katz testing.** N = 6139.

| Kato-Katz Indicator | Any STH | Hookworm | *Ascaris lumbricoides* | *Trichuris trichiura* |
|---|---|---|---|---|
| UNWEIGHTED ESTIMATES | | | | |
| **Kato-Katz tests outcomes: number of participants (prevalence[%])[1]** | | | | |
| Positive | 324 (5.3) | 199 (3.2) | 126 (2.0) | 5 (0.1) |
| Negative | 5,815 (94.7) | 5,940 (96.8) | 6,013 (98.0) | 6,134 (99.9) |
| **Intensity of infection, among positive Kato-Katz tests: number of participants (prevalence [%])[2]** | | | | |
| Light-intensity | 258 (79.6) | 189 (95.0) | 71 (56.3) | 4 (80.0) |
| Moderate-intensity | 54 (16.7.) | 4 (2.0) | 50 (39.7) | 0 (0.0) |
| Heavy-intensity | 12 (3.7) | 6 (3.0) | 5 (4.0) | 1 (20.0) |
| **Unweighted prevalence of moderate/heavy intensity infections: number of participants (prevalence[%])** | | | | |
| Moderate- to-Heavy-intensity infection | 66 (0.2) | 10 (0.0) | 55 (0.9) | 1 (0) |

Notes

[1] Positivity was defined as the presence of eggs on one of two slides read by laboratory technicians.

[2] Light-intensity infections are defined as 1–4,999 epg of feces for *Ascaris lumbricoides* infection, 1–999 epg for *Trichuris trichiura* and 1–1,999 epg for Hookworms. Moderate-intensity infections are defined as 5,000–49,999 epg for *Ascaris lumbricoides*, 1,000–9,999 epg for *Trichuris trichiura* and 2,000–3,999 epg for Hookworms. Heavy-intensity infections are defined as >50,000 epg for *Ascaris lumbricoides*, >10,000 epg for *Trichuris trichiura* and >4,000 epg for Hookworms.

infected with hookworm and *Ascaris lumbricoides*. Due to the small number of *Trichuris trichiura* infections, only the analyses focused on hookworm and *Ascaris lumbricoides* are presented. Among all infections, 258 (79.7%) were light-intensity, 54 (16.7%) were moderate-intensity and 12 (3.7%) were heavy intensity infections. Hookworm was significantly more prevalent in adults than in SAC or PSAC (4.4% *versus* 2% *versus* 1% respectively, Chi$^2$, p<0.001). SAC were significantly more infected with *Ascaris lumbricoides* compared to PSAC or adults (3.0% *versus* 2.0% *versus* 1.7% respectively, Chi$^2$, p = 0.02) (Fig 3). A higher proportion of males than females was infected with hookworm (4% *versus* 2.6%; p = 0.002) and *Ascaris lumbricoides* (2.6% *versus* 1.6%; p = 0.004).

## Intensity of STH infection

In Kato-Katz positive samples, the median egg density for hookworm was 108 EPG (IQR: 48–312), 3,840 EPG (IQR: 312–15,180) for *Ascaris lumbricoides* and 120 EPG (IQR: 60–468) for *Trichuris trichiura*. The intensity of infection was similar in all age groups for hookworm (ANOVA, p = 0.22), with a median egg density of 264 EPG (IQR: 36–384) in PSAC, 96 EPG (IQR: 24–312) in SAC and 108 EPG (IQR: 48–288) in adults. We found a difference in intensity of infection with *Ascaris lumbricoides* between age-groups (ANOVA, p = 0.005), this difference was between SAC and adults (Bonferroni, p = 0.004). Median egg densities were 6,972 EPG (IQR: 264–26292) for PSAC, 7,848 EPG (IQR: 3,714–25,314) for SAC and 780 EPG (IQR: 36–8,772) for adults.

Moderate-to-heavy intensity (MHI) infections were found in 66 out of 6,139 individuals overall (1.1%), amongst whom 10 (0.2%) had MHI infections with hookworm spp., 55 (0.9%) had MHI infections with *Ascaris lumbricoides* and 1 (<0.1%) had MHI infections with *Trichuris trichiura* (Table 2). The burden of MHI infections was greatest in SAC with 2.1% (25/1,184) prevalence of MHI infections of *Ascaris lumbricoides*. (S2 Table) 68.2% (45/66) of MHI infections were found in males (S3 Table). MHI infections were distributed in 15/40 clusters. MHI infections with hookworm were present in 7/40 clusters, MHI infections with *Ascaris lumbricoides* in 7/40 clusters and MHI infections with *Trichuris trichiura* in 1 cluster. There were two clusters showing a particularly high burden of *Ascaris lumbricoides*, with respectively 19 (12.7%) and 30 (20%) individuals with MHI with *Ascaris lumbricoides*.

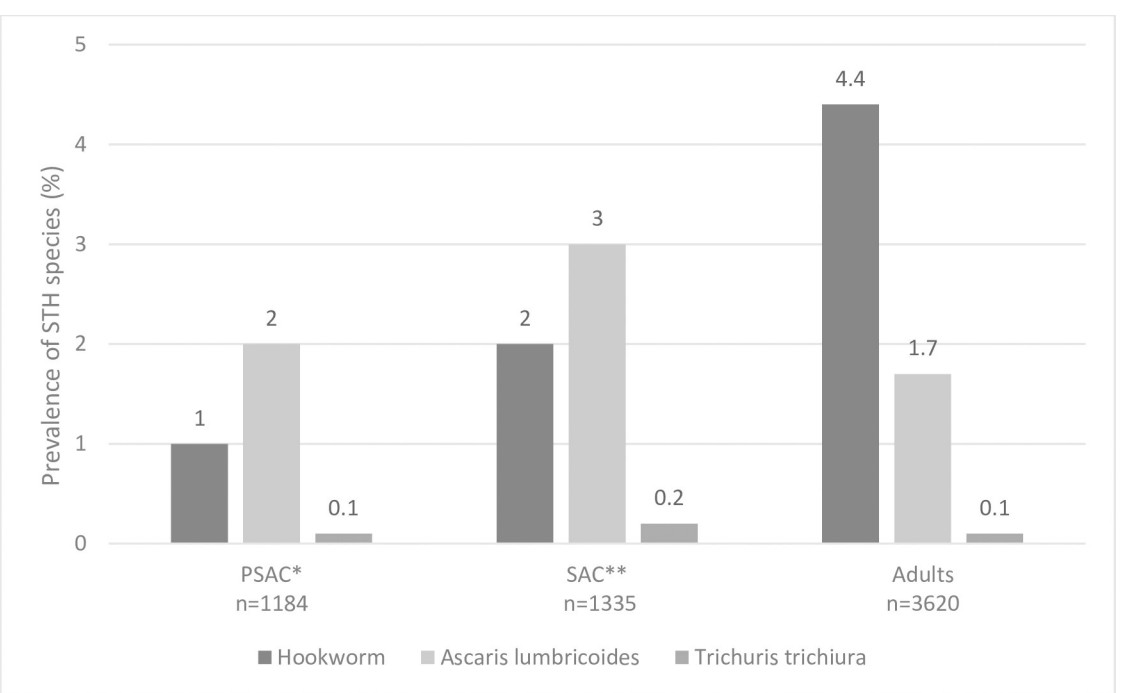

**Fig 3. Soil Transmitted Helminths unweighted prevalence across age-groups.** *PSAC: Pre-School Aged Children, **SAC: School Aged Children.

### Age- and sex-related prevalence and intensity of STH infection (hookworm and *Ascaris lumbricoides*)

Figs 4 and 5 show the age-infection profile for hookworm and *Ascaris lumbricoides*, respectively. The prevalence of hookworm increased with age in both sexes. The prevalence was similar in males and females among PSAC and SAC. However, in adults the prevalence in males was higher than in females except for 50–60 year olds, in whom females were more frequently infected. The intensity of hookworm infection was similar in males and females regardless of age, and was higher in adults than in children. The prevalence of *Ascaris lumbricoides* infection was similar in males and females across all ages, with the period of adolescence and early adulthood (between 10 and 18 years old) corresponding to the period with highest prevalence of *Ascaris lumbricoides* infection in males and the lowest in females (6% for males *versus* 1% for females). The intensity of *Ascaris lumbricoides* infection followed the same profile as prevalence in both sexes.

### Community-level correlation between intensity and prevalence of STH infection

We found a positive linear relationship between STH infection prevalence and the intensity of infection at cluster level in our study population (Fig 6). This correlation was strong for both hookworm (r = 0.73, p<0.0001) and *Ascaris lumbricoides* (r = 0.98, p<0.0001).

### Factors associated with hookworm infection

The results of univariate analyses of factors associated with hookworm infection prevalence are presented in S4 Table. In this section, we present the results of multivariable analyses are presented (Table 3).

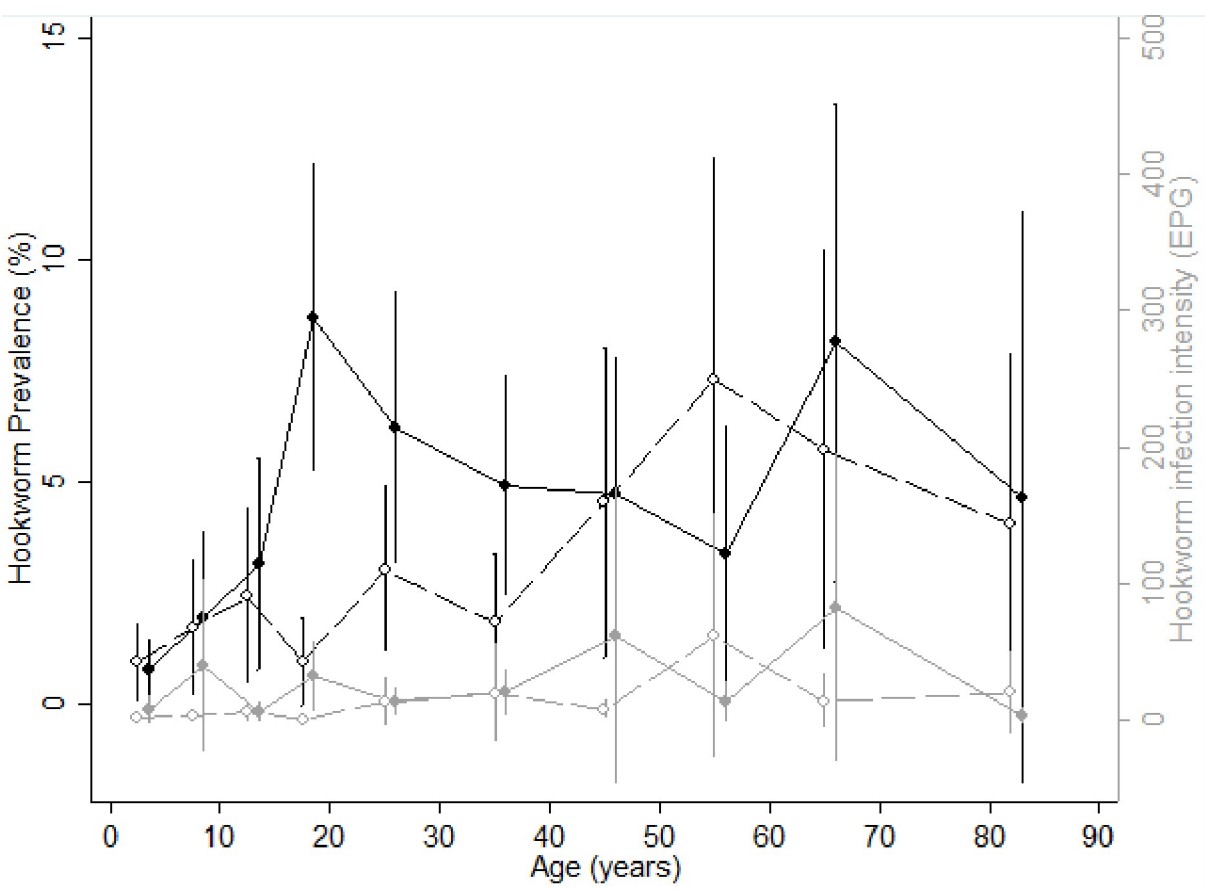

**Fig 4. Age-infection profiles for hookworm.** (A) Prevalence (black lines) and intensity (grey lines) of hookworm infection by age for males (solid line and circles) and females (dashed lines and empty circles). Vertical bars represent confidence intervals.

At an individual level, PSAC and SAC were significantly less likely to be infected with hookworm than adults (aOR = 0.2, 95%CI 0.1–0.4, p< 0.001 and aOR = 0.5, 95%CI 0.3–0.8, p = 0.008, respectively). Females were also significantly less likely to be infected than males (aOR = 0.6, 95%CI 0.4–0.8, p = 0.004). Individuals who reported a history of deworming during the past year were significantly less likely to be infected (aOR = 0.4, 95%CI 0.3–0.7, p< 0.002).

Among household factors, the household asset index, a proxy measure for family wealth, showed that individuals in the poorest households had a significantly higher odds of infection than the richest (5th quintile) with a significant dose-response effect (Cuzick test of trend, p<0.001), (First quintile: aOR = 5.0, 95%CI 2.1–12.0, p<0.001, 2nd quintile aOR = 3.6, 95%CI 1.5–8.7, p = 0.001 and 3rd quintile aOR = 2.5, 95%CI 1.0–6.0, p = 0.02). With respect to occupational exposure, farmers were more likely to be infected with hookworm than others (aOR = 1.8, 95%CI 1.1–2.9, p = 0.02). Individuals living in medium population density settings were more likely to be infected than those living in high density settings, (aOR = 2.6, 95%CI 1.2–5.4, p = 0.01). (Table 3)

Among WASH factors, household sanitation and open defecation were found to be strongly associated with hookworm infection. Individuals using improved unshared sanitation facilities had half the odds of hookworm infection compared to those defecating outdoors (aOR = 0.5, 95%CI 0.2–0.9, p = 0.04). (Table 3)

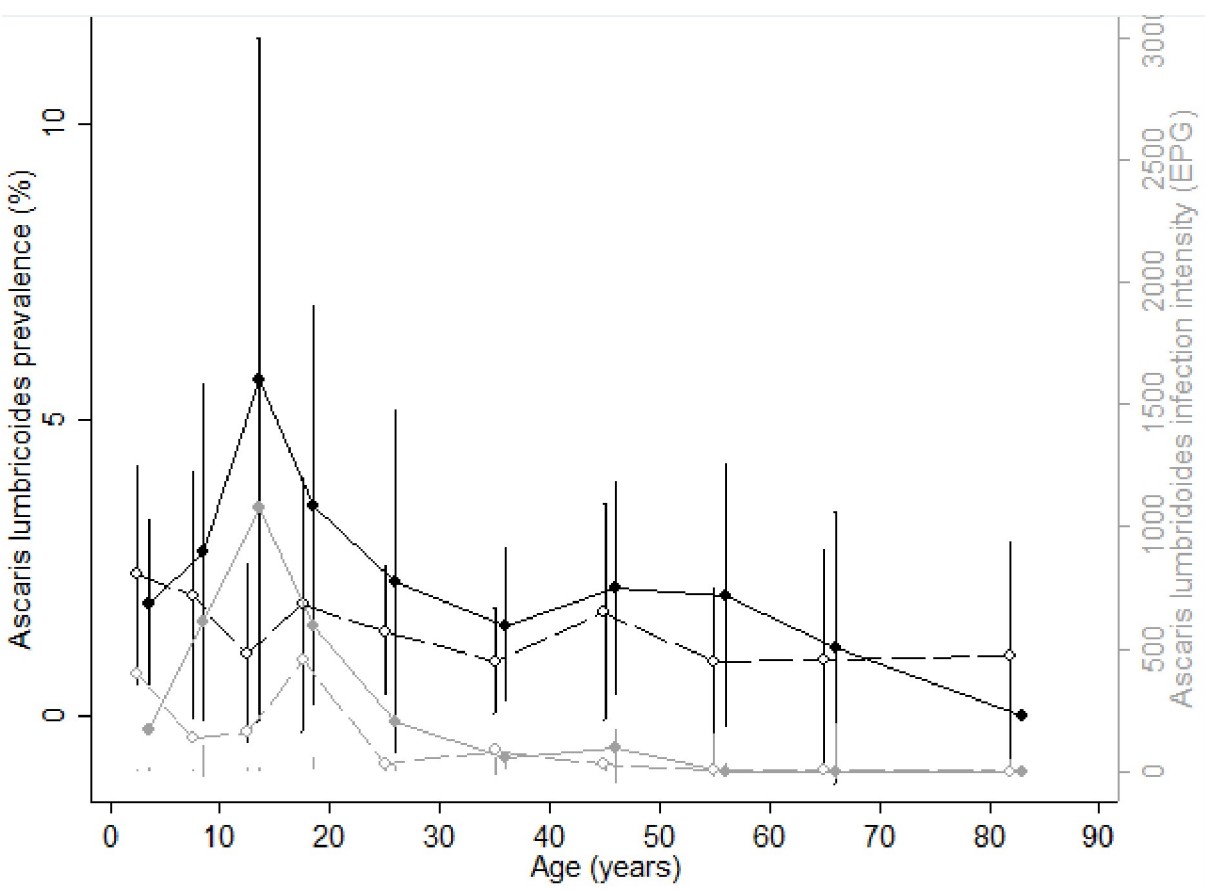

**Fig 5. Age-infection profiles for *Ascaris lumbricoides*.** (A) Prevalence (black lines) and intensity (grey lines) of *Ascaris* infection by age for males (solid line and circles) and females (dashed lines and empty circles). Vertical bars represent confidence intervals.

### Factors associated with intensity of hookworm infection

Children had a significantly lower intensity of hookworm infection as compared to adults (PSAC: adjusted IIR = 0.1, 95%CI 0.0–0.3, p<0.001; SAC: adjusted IIR = 0.3, 95%CI 0.1–0.7, p = 0.01). Females had significantly lower intensity infections than males (adjusted IIR = 0.3, 95%CI 0.2–0.6, p = 0.001), as did individuals dewormed the year before (adjusted IIR = 0.2, 95%CI 0.1–0.5, p<0.001). (Table 3)

At the household level, less educated people (those with no education, primary school or secondary school) had higher intensity infections with hookworm compared to those with university level education (adjusted IIR = 40.1, 95%CI 2.5–652.8, p = 0.01; adjusted IIR = 30.9, 95%CI 1.9–513.9, p = 0.02; adjusted IIR = 19.3, 95%CI 1.2–308.8, p = 0.04, respectively). Being a farmer, living in a house with natural floor material versus man-made floor material, and living in a peri-urban setting were also all factors found to be associated with a significantly increased intensity of hookworm infections (farmer: adjusted IIR = 3.9, 95%CI 1.7–9.3, p = 0.002; natural floor material: adjusted IIR = 0.2, 95%CI 0.0–1.0, p = 0.06; peri-urban settings: adjusted IIR = 6.2, 95%CI 1.8–20.9, p = 0.003).(Table 3)

Access to unimproved water available more than 30 minutes away from the house was associated with significantly higher intensity hookworm infection (adjusted IIR = 13.5, 95%CI 1.6–111.5; p = 0.02) compared to improved water available less than 30 minutes from the house. Compared to open defecation behavior, using an improved and unshared toilet was associated

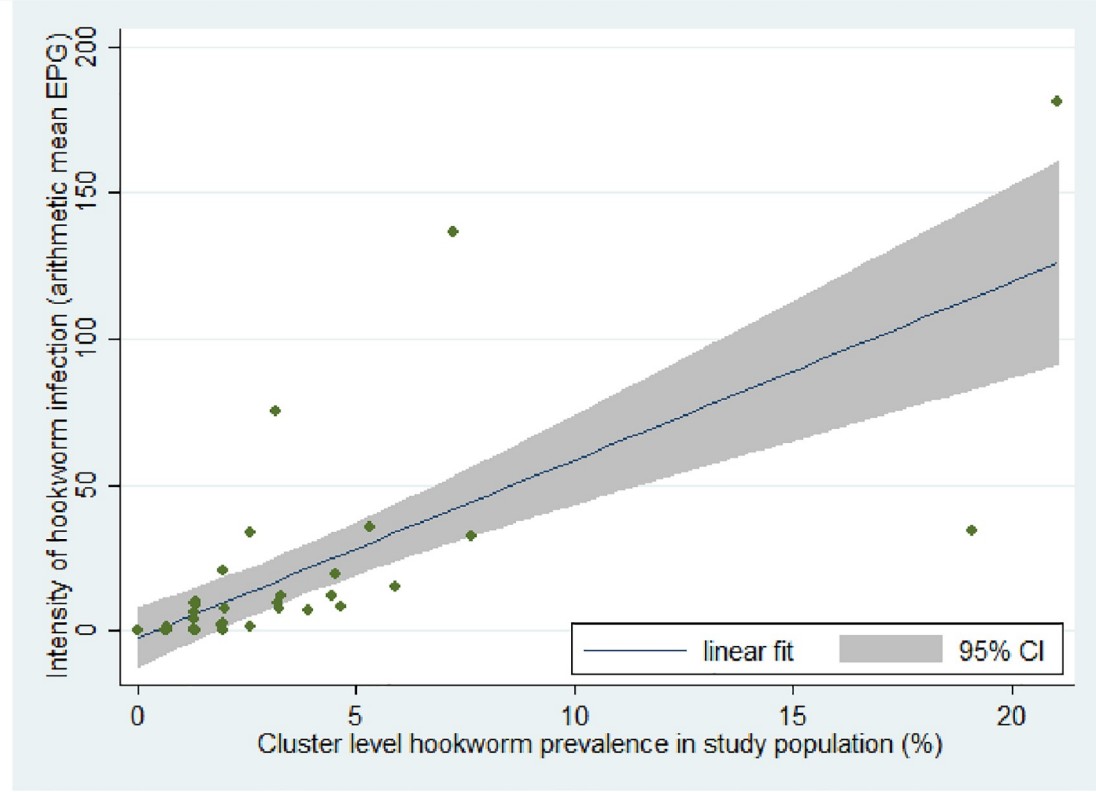

(A)

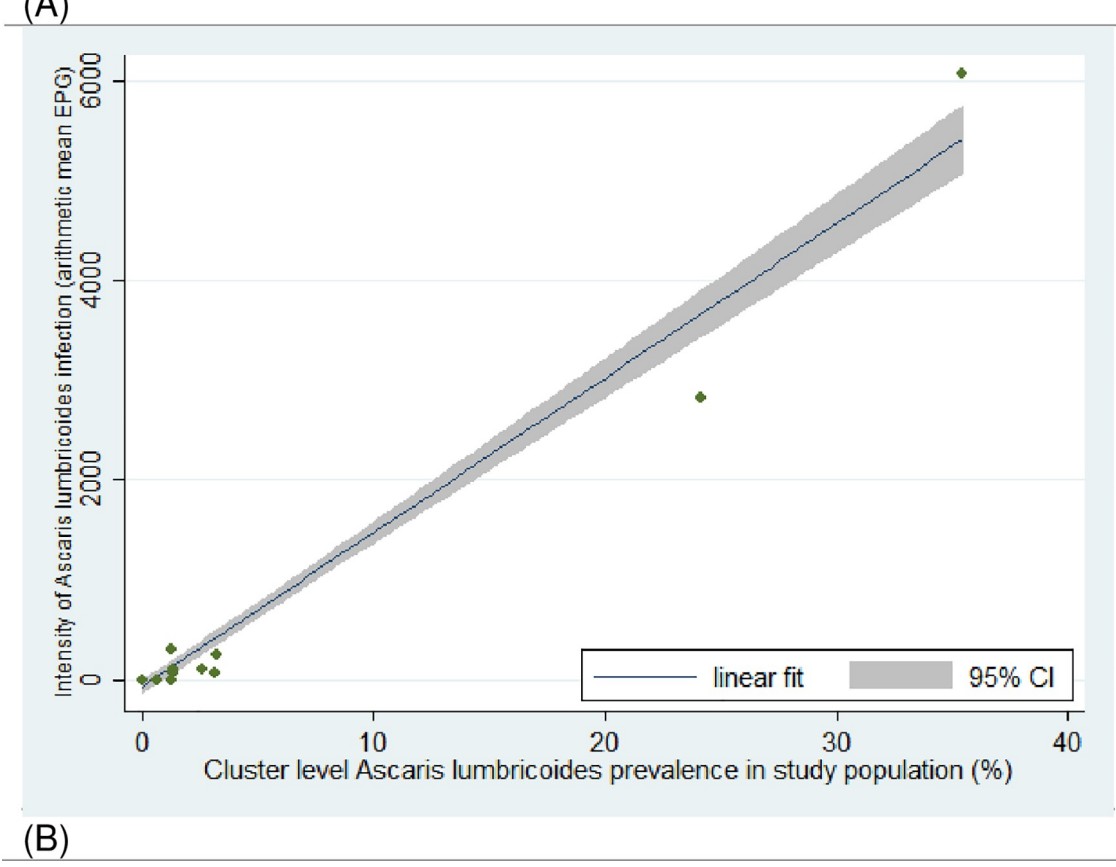

(B)

**Fig 6. Cluster level correlation between prevalence and intensity of hookworm and *Ascaris lumbricoides* infection in the study population.** (A) Hookworm infection in study population; (B) *Ascaris lumbricoides* infection in study population.

with significantly lower intensity hookworm infections (adjusted IIR = 0.2, 95%CI 0.1–0.7, p = 0.01). (Table 3) No environmental factor was found to be associated with intensity of hookworm infections in multivariable analyses.

### Factors associated with *Ascaris lumbricoides* infection prevalence

Among the individual factors assessed, SAC (5–14 years) were significantly more likely to be infected with *Ascaris lumbricoides* than adults (aOR = 2.0, 95%CI 1.1–3.6, p = 0.01). However, no difference in odds of infection was found between PSAC and adults. Female individuals were less likely to be infected with *Ascaris lumbricoides* than males (aOR = 0.5, 95%CI 0.3–0.9, p = 0.02). (Table 4)

Amongst environmental factors, low soil acidity was significantly associated with increased odds of *Ascaris lumbricoides* infection compared to the highest soil acidity (aOR = 4.8, 95%CI 1.8–13.1, p = 0.002). Moderate [29.6–31.9°C] and high [31.9; 32.8°C] daytime land surface temperatures were associated with lower odds of infection with *Ascaris lumbricoides* compared to lower temperatures [26.2–29.6°C [(aOR = 0.12, 95%CI 0.0–0.4, p = 0.001; and aOR = 0.17, 95%CI 0.0–0.9, p = 0.04 respectively). The summary of the multivariable analysis with *Ascaris lumbricoides* is presented in Table 4.

### Intra-Class Correlation statistics for hookworm and *Ascaris lumbricoides* infection prevalence

Comparison of the Intra-Class Correlation (ICC) values between models containing only age and sex (model 1) and the fully adjusted multivariable final model with all the fixed effect covariables (model 2) showed decreased ICC values in the fully adjusted model, considering either level-3 ICC at the cluster level or level-2 ICC at the household-within-cluster level (S5 Table).

When only adjusting for age and sex, the prevalence of hookworm infection was correlated between individuals within the same cluster (ICC = 0.16, 95%CI 0.10–0.26), and this correlation increased significantly between individuals within the same household and cluster level (ICC = 0.58, 95%CI 0.40–0.74). Prevalence of *Ascaris lumbricoides* infection was moderately correlated within the same cluster (ICC = 0.54, 95%CI 0.34–0.73), with a small increase within the same household and cluster level (ICC = 0.60, 95%CI 0.26–0.76). In this model, household and cluster random effects compose approximately 58% and 60% of the total residual variance for hookworm and *Ascaris lumbricoides* infection prevalence, respectively.

In the fully adjusted model, we found a correlation of hookworm infection prevalence within the same cluster (ICC = 0.03, 95%CI 0.01–0.10), although this correlation increased within the same household and cluster level (ICC = 0.39, 95%CI 0.17–0.65). *Ascaris lumbricoides* infection prevalence was moderately correlated between individuals within the same cluster (ICC = 0.42, 95%CI 0.23–0.64), and this correlation increased slightly within the same household and cluster level (ICC = 0.51, 95%CI 0.26–0.76). We estimated that household and cluster random effects compose approximately 39% and 51% of the total residual variance of hookworm and *Ascaris lumbricoides* infection prevalence, respectively.

### Discussion

We observed a relatively low prevalence of STHs in this region of Benin. Hookworm were the most prevalent infections, and were more prevalent in adults, while *Ascaris lumbricoides* was

**Table 3. Factors associated with the prevalence and the intensity of hookworm infection in Comé, Bénin: findings from a cross-sectional baseline prevalence survey in the DeWorm3 STH–elimination trial.**

| Variables | Number of participants with hookworm infection (prevalence [%]) | Intensity of hookworm infection (EPG) | Generalized Logistic Mixed Model Multivariate Analysis †§ | | Negative binomial regression Multivariate Analysis ‡♣ | |
|---|---|---|---|---|---|---|
| | | median (IQR), [min, max] | Adjusted Odds Ratio (95%CI) | p-value | Adjusted Infection Intensity Ratio (95% CI) | p-value |
| INDIVIDUAL factors | | | | | | |
| Age | n = 6,139 | n = 6,138 | | <0.0001 | | |
| - Adults (≥15 years) | 160 (4.4) | 0(0–0), [0–12,960] | Reference | | Reference | |
| - PreSAC (1–4 years) | 27 (2.0) | 0 (0–0), [0–3,552] | 0.2 (0.1–0.4) | < 0.001 | 0.1 (0.0–0.3) | <0.001 |
| - SAC (5–14 years) | 12 (1.0) | 0 (0–0), [0–11,100] | 0.5 (0.3–0.8) | 0.008 | 0.3 (0.1–0.7) | 0.01 |
| Gender | n = 6,139 | n = 6,138 | | 0.004 | | |
| - Male | 113 (4.0) | 0 (0–0), [0–12,960] | Reference | | Reference | |
| - Female | 86 (2.6) | 0 (0–0), [0–9,840] | 0.6 (0.4–0.8) | 0.004 | 0.3 (0.2–0.6) | 0.001 |
| History of deworming during the past year | n = 6,091 | n = 6,090 | | 0.002 | | |
| - No | 171 (4.6) | 0 (0–0), [0–12,960] | Reference | | Reference | |
| - Yes | 28 (1.2) | 0 (0–0), [0–3,48] | 0.5 (0.3–0.7) | < 0.002 | 0.2 (0.1–0.5) | <0.001 |
| Shoe wearing behavior | n = 6,091 | n = 6,090 | * | | * | |
| - Shoes | 100 (3.0) | 0 (0–0), [0–12,960] | | | | |
| - No shoes | 99 (3.6) | 0 (0–0), [0–11,100] | | | | |
| Current school attendance | n = 6,139 | n = 6,138 | * | | * | |
| - No | 152 (3.5) | 0(0–0), [0–12,960] | | | | |
| - Yes | 47 (2.5) | 0(0–0), [0–11,100] | | | | |
| HOUSEHOLD factors | | | | | | |
| Highest education level in the household | n = 6,139 | n = 6,138 | * | | | |
| - University/College/Diploma | 4 (0.6) | 0(0–0), [0–228] | | | Reference | |
| - No education | 93 (4.8) | 0(0–0), [0–12,960] | | | 40.1 (2.5–652.8) | 0.01 |
| - Primary | 50 (3.7) | 0(0–0), [0–11,100] | | | 30.9 (1.8–513.9) | 0.02 |
| - Secondary | 52 (2.4) | 0(0–0), [0–4,764] | | | 19.3 (1.2–308.8) | 0.04 |
| Quintiles of household asset index | n = 6,139 | n = 6,138 | | <0.0001 | * | |
| - 5th quintile (richest) | 16 (1.0) | 0(0–0), [0–1,440] | Reference | | | |
| - 1st quintile (poorest) | 76 (7.7) | 0(0–0), [0–12,960] | 5.0 (2.1–12.0) | <0.001 | | |
| - 2nd quintile | 51 (4.9) | 0(0–0), [0–4,764] | 3.6 (1.5–8.7) | 0.001 | | |
| - 3rd quintile | 38 (3.2) | 0(0–0) [0–9,840] | 2.5 (1.0–6.0) | 0.02 | | |
| - 4th quintile | 18 (1.3) | 0(0–0), [0–1,104] | 0.9(0.4–2.5) | 0.91 | | |
| Head of household's occupation | n = 6,139 | n = 6,138 | | 0.03 | | |
| - Others/ Don't know/Refused | 98 (2.1) | 0(0–0), [0–4,764] | Reference | | Reference | |
| - Farmer | 89 (9.7) | 0(0–0), [0–12,960] | 1.8 (1.1–2.9) | 0.02 | 3.9 (1.7–9.3) | 0.002 |
| - Fisher | 12 (2.0) | 0(0–0), [0–516] | 0.7 (0.3–1.6) | 0.70 | 0.2 (0.0–1.0) | 0.06 |
| Observed floor type: natural/manmade | n = 6,139 | n = 6,138 | * | | | |
| - Man-made floor material | 123 (2.4) | 0(0–0), [0–11,100] | | | Reference | |
| - Natural floor material | 75 (7.9) | 0(0–0),[0–12,960] | | | 3.0 (1.4–6.7) | 0.01 |
| - Other/Don't know/Refused | 1 (3.7) | 0(0–0), [0–48] | | | 3.5 (0.0–664.6) | 0.65 |
| Urbanization | n = 6,134 | n = 6,133 | | 0.02 | | |
| - Urban | 29 (1.2) | 0(0–0), [0–9,840] | Reference | | Reference | |
| - Peri-urban | 146 (5.0) | 0(0–0), [0–12,960] | 2.6 (1.2–5.4) | 0.01 | 6.2 (1.8–20.9) | 0.003 |
| - Rural | 24 (3.0) | 0(0–0), [0–3,120] | 1.4 (0.6–3.2) | 0.48 | 1.9 (0.4–8.5) | 0.41 |
| 3 tertiles of population density at 1km² | n = 6,134 | n = 6,133 | * | | * | |

(Continued)

**Table 3.** (Continued)

| Variables | Number of participants with hookworm infection (prevalence [%]) | Intensity of hookworm infection (EPG) | Generalized Logistic Mixed Model Multivariate Analysis †§ | | Negative binomial regression Multivariate Analysis ‡♣ | |
|---|---|---|---|---|---|---|
| | | median (IQR), [min, max] | Adjusted Odds Ratio (95%CI) | p-value | Adjusted Infection Intensity Ratio (95% CI) | p-value |
| 1st tertile [3; 542 [low | 135 (6.7) | 0(0–0) [0–12,960] | | | | |
| 2nd tertile [542; 1235[medium | 43 (2.1) | 0(0–0), [0–3,624] | | | | |
| 3rd tertile [1235; 2528] high | 21 (1.0) | 0(0–0), [0–9,840] | | | | |
| WASH factors | | | | | | |
| Household water SDG service modified | n = 6,135 | n = 6,134 | * | | | |
| - Improved ≤ 30min | 141 (2.8) | 0 (0–0),[0–2,124] | | | Reference | |
| - Surface water > 30min | 0 (0.0) | 0(0–0), [0–3120] | | | 0 | - |
| - Surface water ≤ 30min | 2 (12.5) | 0(0–0), [0–120] | | | 45.6 (0.8–2726.5) | 0.07 |
| - Unimproved > 30min | 6 (11.3) | 0(0–0), [0–696] | | | 13.5 (1.6–111.5) | 0.02 |
| - Unimproved ≤ 30min | 39 (6.7) | 0(0–0), [0–2,124] | | | 1.9 (0.7–5.1) | 0.21 |
| - Improved > 30 min | 11 (2.9) | 0(0–0), [0–3,120] | | | 0.7 (0.2–3.0) | 0.62 |
| | | | | 0.24 | | |
| Household sanitation SDG service | n = 5,816 | n = 5,815 | Reference | | | |
| - Open defecation | 129 (6.0) | 0(0–0), [0–12,960] | 0.5 (0.1–1.9) | 0.32 | Reference | |
| - Unimproved shared | 3 (1.3) | 0(0–0), [0–60] | 0.2 (0.0–2.2) | 0.20 | 0.2 (0.0–2.3) | 0.19 |
| - Unimproved unshared | 1 (0.8) | 0(0–0), [0–204] | 0.7 (0.4–1.4) | 0.33 | 0.1 (0.002–2.16) | 0.13 |
| - Improved shared | 28 (1.7) | 0(0–0), [50–9,840] | 0.5 (0.2–1.0) | 0.04 | 0.4 (0.116–1.25) | 0.12 |
| - Improved unshared | 28 (1.7) | 0(0–0), [0–1,440] | ** | 0.46 | 0.2 (0.07–0.70) | 0.01 |
| Household Hand washing facility SDG service | n = 5,716 | n = 5,715 | Reference | | ** | |
| - No facility | 64 (3.4) | 0(0–0), [0–12,960] | 0.9 (0.6–1.4) | 0.76 | Reference | |
| - Limited | 104 (3.3) | 0(0–0), [0–8,064] | 0.6 (0.3–1.3) | 0.22 | 0.8 (0.– 1.8) | 0.64 |
| - Basic | 13 (1.8) | 0(0–0), [0–1,440] | | | 0.4 (0.1–1.7) | 0.24 |
| ENVIRONMENTAL Factors | | | | | | |
| Elevation (in meters) | n = 6,134 | 6,133 | ** | 0.12 | * | |
| 1st tertile [-1; 15 [(low) | 63 (3.0) | 0(0–0), [0–12,960] | Reference | | | |
| 2nd tertile [15; 30 [(medium) | 37 (1.6) | 0(0–0), [0–9,840] | 1.1 (0.5–2.2) | 0.85 | | |
| 3rd tertile [30; 61) (high) | 99 (5.7) | 0(0–0), [0–11,100] | 1.7 (1.0–3.1) | 0.06 | | |
| Proportion of soil that is sand at the surface at 0 cm (%) | n = 6,134 | n = 6,133 | * | | * | |
| 1st tertile [35; 55 [(low) | 37 (1.7) | 0(0–0), [0–1,176] | | | | |
| 2nd tertile [55; 64 [(medium) | 40 (2.0) | 0(0–0), [0–3,624] | | | | |
| 3rd tertile [64; 78) (high) | 122 (6.1) | 0(0–0), [0–12,960] | | | | |
| Soil acidity (pH KCL) at everage depth (0-5-15 cm) | n = 6,134 | n = 6,133 | * | | * | |
| 1st tertile [4.8; 5.1 [(low) | 40 (2.0) | 0(0–0), [0–11,100] | | | | |
| 2nd tertile [5.1; 5.2 [(medium) | 90 (4.0) | 0(0–0), [0–9,840] | | | | |
| 3rd tertile [5.2; 5.7) (high) | 69 (3.7) | 0(0–0), [0–12,960] | | | | |
| MODIS daytime land surface temperature mean for 2018 (˚celsius) | n = 6,134 | n = 6,133 | * | | * | |
| 1st tertile [26.2; 29.6 [(low) | 71 (3.4) | 0(0–0), [0–11,100] | | | | |
| 2nd tertile [29.6; 31.9 [(medium) | 109 (4.5) | 0(0–0), [0–12,960] | | | | |
| 3rd tertile [31.9; 32.8) (high) | 19 (1.2) | 0(0–0), [0–3,624] | | | | |

(Continued)

**Table 3.** (Continued)

| Variables | Number of participants with hookworm infection (prevalence [%]) | Intensity of hookworm infection (EPG) | Generalized Logistic Mixed Model Multivariate Analysis †§ | | Negative binomial regression Multivariate Analysis ‡♣ | |
|---|---|---|---|---|---|---|
| | | median (IQR), [min, max] | Adjusted Odds Ratio (95%CI) | p-value | Adjusted Infection Intensity Ratio (95% CI) | p-value |
| MODIS Enhanced Vegetation Index (EVI) mean for 2018 | n = 6,134 | n = 6,133 | * | | * | |
| 1st tertile [0.04; 0.2 [(low) | 26 (1.2) | (0–0), [0–9,840] | | | | |
| 2nd tertile [0.2; 0.3 [(medium) | 35 (1.7) | (0–0), [0–3,624] | | | | |
| 3rd tertile [0.3; 0.4] (high) | 138 (6.9) | (0–0), [0–12,960] | | | | |
| MODIS normalized difference vegetation index (NDVI) mean for 2018 | n = 6,134 | n = 6,133 | ** | 0.007 | ** | |
| 1st tertile [0.06; 0.3 [(low) | 27 (1.3) | (0–0), [0–9,840] | Reference | | Reference | |
| 2nd tertile [0.3; 0.4 [(medium) | 33 (1.6) | (0–0), [0–3,624] | 0.7 (0.3–1.6) | 0.41 | 0.4 (0.1–1.6) | 0.21 |
| 3rd tertile [0.4; 0.6] (high) | 139 (7.0) | (0–0), [0–12,960] | 2.0 (0.9–4.3) | 0.07 | 3.3 (0.8–12.9) | 0.08 |
| Aridity index | n = 6,134 | n = 6,133 | * | | * | |
| 1st tertile [0.59; 0.61 [(low) | 81 (3.9) | (0–0), [0–9,840] | | | | |
| 2nd tertile [0.61; 0.62 [(medium) | 50 (2.4) | (0–0), [0–8,064] | | | | |
| 3rd tertile [0.62; 0.65] (high) | 68 (3.5) | (0–0), [0–12,960] | | | | |

Notes

† Adjusted Generalized logistic mixed model estimating equations with exchangeable correlation structure.

§ 5,366 observations included in fully adjusted model.

‡ Adjusted zero-inflated negative binomial regression model, inflating for sex and age (1–4 years, 5–14 years, 15 years), with an exchangeable correlation matrix.

♣ 5,364 observations included in fully adjusted model.

* Variable dropped from fully adjusted model during model adjustment process using lowest AIC criteria.

** Variable in the final adjusted model but with no significant category

Abbreviation: School Aged Children (SAC), Pre School Aged Children (PSAC), confidence interval (CI), interquartile range (IQR), Moderate Resolution Imaging Spectroradiometer (MODIS)

more prevalent in children. Females were generally less infected than males across all ages. Females, children, those dewormed during the previous year and those using improved unshared sanitation facilities had lower odds of hookworm infections, while being a farmer, living in peri-urban settings versus urban and being poor was associated with a higher odds of hookworm infection. In addition to those factors, the intensity of hookworm infection also decreased if an improved water source was available at less than 30 minutes walking distance.

Since 2013, the Ministry of Health in Benin has focused its efforts on developing and implementing strategies for the control of five NTDs considered to be of highest priority, namely trachoma, onchocerciasis, lymphatic filariasis, schistosomiasis and STHs. Those efforts were bolstered markedly through the ENVISION program [25], a U.S. Agency for International Development (USAID)-funded initiative that ran from 2013 through 2019 in Benin. A nationwide STH prevalence survey that was completed in 2015, that reported 20% prevalence [13] in SAC in the Comé district. Following that national mapping effort, three rounds of school-based MDA with albendazole were undertaken according to the recommendations of WHO, i.e. primarily targeting SAC and PSAC for either once or twice yearly treatment as a function of the estimated prevalence of infection in any given district. [26] Coverage of SAC with school MDA between 2015 and 2017 was estimated between 59% and 83%. Albendazole and/or mebendazole are also distributed in health facilities and to pregnant women during routine

**Table 4. Factors associated with the prevalence of *Ascaris lumbricoides* infection in Comé, Bénin: findings from a baseline prevalence survey using Kato-Katz technique.**

| Variables | Number of participants with *Ascaris lumbricoides* infection (prevalence [%]) | Generalized Logistic Mixed Model Univariate Analysis | | Generalized Logistic Mixed Model Multivariate Analysis † | |
|---|---|---|---|---|---|
| | | Odds Ratio (95% CI) | p-value | Adjusted Odds Ratio (95%CI) | p-value |
| INDIVIDUAL factors | | | | | |
| Age | n = 6,139 | | 0.02 | | |
| - Adults (≥15 years) | 62 (1.71) | Reference | | Reference | |
| - PreSAC (1–4 years) | 24 (2.03) | 1.3 (0.7–2.2) | 0.42 | 1.6 (0.8–3.1) | 0.14 |
| - SAC (5–14 years) | 40 (3.00) | 2.0 (1.2–3.3) | 0.005 | 2.0 (1.1–3.6) | 0.01 |
| Gender | n = 6,139 | | | | |
| - Male | 74 (2.62) | Reference | | Reference | |
| - Female | 52 (1.57) | 0.5 (0.3–0.8) | 0.003 | 0.5 (0.3–0.9) | 0.02 |
| History of deworming during the past year | n = 6,091 | | | * | |
| - No | 98 (2.66) | Reference | | | |
| - Yes | 28 (1.16) | 0.7 (0.4–1.2) | 0.24 | | |
| Shoe wearing behavior | n = 6,091 | | | * | |
| - Shoes | 53 (1.58) | Reference | | | |
| - No shoes | 73 (2.66) | 0.8 (0.5–1.3) | 0.41 | | |
| Current school attendance | n = 6,139 | | | ** | |
| - No | 72 (1.68) | Reference | | | |
| - Yes | 54 (2.93) | 2.0 (1.3–3.0) | 0.001 | | |
| HOUSEHOLD factors | | | | | |
| Highest education level in the household | n = 6,139 | | 0.08 | ** | |
| - University/College/Diploma | 3 (0.43) | Reference | | | |
| - No education | 41 (2.11) | 2.4 (0.6–9.0) | 0.21 | | |
| - Primary | 45 (3.36) | 4.2 (1.1–16.6) | 0.04 | | |
| - Secondary | 37 (1.71) | 3.1 (0.8–11.9) | 0.10 | | |
| Quintiles of household asset index | n = 6,139 | | 0.14 | ** | |
| - 5th quintile (richest) | 13 (0.83) | Reference | | | |
| - 1st quintile (poorest) | 43 (4.37) | 1.8 (0.8–4.1) | 0.15 | | |
| - 2nd quintile | 30 (2.88) | 1.7 (0.7–3.9) | 0.21 | | |
| - 3rd quintile | 22 (1.87) | 0.9 (0.4–2.0) | 0.72 | | |
| - 4th quintile | 18 (1.31) | 0.9 (0.4–2.3) | 0.94 | | |
| Head of household's occupation | n = 6,139 | | 0.21 | * | |
| - Others/ Don't know/Refused | 62 (1.34) | Reference | | | |
| - Farmer | 13 (1.41) | 0.9 (0.4–1.9) | 0.86 | | |
| - Fisher | 51 (8.50) | 1.6 (0.9–2.6) | 0.10 | | |
| Observed floor type: natural/manmade | n = 6,139 | | 0.99 | * | |
| - Man-made floor material | 97 (1.88) | Reference | | | |
| - Natural floor material | 29 (3.05) | 1.0 (0.6–1.7) | 0.99 | | |
| - Other/Don't know/Refused | 1 (3.7) | - | - | | |
| Urbanization | n = 6,134 | | 0.26 | * | |
| - Urban | 69 (2.85) | Reference | | | |
| - Peri-urban | 14 (0.48) | 0.4 (0.1–1.2) | 0.11 | | |
| - Rural | 43 (5.42) | 0.8 (0.5–1.4) | 0.51 | | |
| 3 tertiles of population density at 1km | n = 6,134 | | 0.005 | * | |
| - 1st tertile [3; 542 [(low) | 19 (0.94) | Reference | | | |

*(Continued)*

**Table 4.** (Continued)

| Variables | Number of participants with *Ascaris lumbricoides* infection (prevalence [%]) | Generalized Logistic Mixed Model Univariate Analysis | | Generalized Logistic Mixed Model Multivariate Analysis † | |
|---|---|---|---|---|---|
| | | Odds Ratio (95% CI) | p-value | Adjusted Odds Ratio (95%CI) | p-value |
| - 2nd tertile [542; 1235 [(medium) | 57 (2.75) | 1.6 (0.8–3.0) | 0.14 | | |
| - 3rd tertile [1235; 2528) (high) | 50 (2.45) | 2.9 (1.5–5.9) | 0.002 | | |
| WASH factors | | | | | |
| Household water SDG service modified | n = 6,063 | | 0.97 | * | |
| - Improved ≤ 30min | 107 (2.10) | Reference | | | |
| - Surface water > 30min | 0 (0.0) | 1 | | | |
| - Surface water ≤ 30min | 0 (0.0) | 1 | | | |
| - Unimproved > 30min | 0 (0.0) | 1 | | | |
| - Unimproved ≤ 30min | 8 (1.37) | 1.1 (0.4–2.7) | 0.85 | | |
| - Improved > 30 min | 11 (2.88) | 1.1 (0.5–2.4) | 0.84 | | |
| - | | | | | |
| Household sanitation SDG service | n = 5,816 | | 0.65 | ** | |
| - Open defecation | 84 (3.89) | Reference | | Reference | |
| - Unimproved shared | 5 (2.20) | 1.3 (0.4–4.3) | 0.65 | 1.6 (0.4–6.8) | 0.51 |
| - Unimproved unshared | 2 (1.57) | 1.2 (0.2–6.9) | 0.84 | 1.1 (0.1–11.5) | 0.94 |
| - Improved shared | 23 (1.38) | 0.9 (0.5–1.7) | 0.74 | 1.1 (0.5–2.1) | 0.87 |
| - Improved unshared | 11 (0.67) | 0.6 (0.3–1.2) | 0.16 | 0.8 (0.3–1.8) | 0.55 |
| Household Hand washing facility SDG service | n = 5,716 | | 0.09 | ** | |
| - No facility | 13 (1.80) | Reference | | Ref | |
| - Limited | 16 (0.85) | 2.6 (1.2–5.6) | 0.01 | 1.8 (0.8–3.6) | 0.13 |
| - Basic | 71 (2.28) | 1.9 (0.7–4.8) | 0.20 | 1.3 (0.5–3.4) | 0.54 |
| ENVIRONMENTAL Factors | | | | | |
| Elevation (in meters) | n = 6,134 | | 0.02 | * | |
| 1st tertile [-1; 15 [(low) | 88 (4.20) | Reference | | | |
| 2nd tertile [15; 30 [(medium) | 25 (1.09) | 0.6 (0.3–1.2) | 0.17 | | |
| 3rd tertile [30; 61) (high) | 13 (0.75) | 0.3 (0.1–0.7) | 0.01 | | |
| Proportion of soil that is sand at the surface at 0 cm (%) | n = 6,134 | | 0.09 | * | |
| 1st tertile [35; 55 [(low) | 101 (4.71) | Reference | | | |
| 2nd tertile [55; 64 [(medium) | 9 (0.45) | 0.5 (0.2–1.1) | 0.10 | | |
| 3rd tertile [64; 78) (high) | 16 (0.80) | 0.5 (0.2–1.1) | 0.07 | | |
| Soil acidity (pH KCL) at average depth (0-5-15 cm) | n = 6,134 | | 0.001 | | |
| 1st tertile [4.8; 5.1 [(low) | 14 (0.69) | Reference | | Reference | |
| 2nd tertile [5.1; 5.2 [(medium) | 29 (1.31) | 2.2 (1.0–5.0) | 0.06 | 2.0 (0.9–4.2) | 0.20 |
| 3rd tertile [5.2; 5.7) (high) | 83 (4.42) | 4.1 (1.9–8.8) | 0.001 | 4.8 (1.8–13.1) | 0.002 |
| MODIS daytime land surface temperature mean for 2018 (˚celsius) | n = 6,134 | | 0.001 | | |
| 1st tertile [26.2; 29.6 [(low) | 115 (5.48) | Reference | | Reference | |
| 2nd tertile [29.6; 31.9 [(medium) | 7 (0.29) | 0.1 (0.03–0.4) | 0.001 | 0.1 (0.0–0.4) | 0.001 |
| 3rd tertile [31.9; 32.8) (high) | 4 (0.25) | 0.1 (0.02–0.5) | 0.005 | 0.2 (0.0–0.9) | 0.038 |
| MODIS Enhanced Vegetation Index (EVI) mean for 2018 | n = 6,134 | | 0.36 | * | |
| 1st tertile [0.04; 0.2 [(low) | 20 (0.96) | Reference | | | |
| 2nd tertile [0.2; 0.3 [(medium) | 67 (3.25) | 1.1 (0.5–2.4) | 0.72 | | |

*(Continued)*

**Table 4.** (Continued)

| Variables | Number of participants with *Ascaris lumbricoides* infection (prevalence [%]) | Generalized Logistic Mixed Model Univariate Analysis | | Generalized Logistic Mixed Model Multivariate Analysis † | |
|---|---|---|---|---|---|
| | | Odds Ratio (95% CI) | p-value | Adjusted Odds Ratio (95%CI) | p-value |
| 3rd tertile [0.3; 0.4] (high) | 39 (1.96) | 0.8 (0.3–1.8) | 0.54 | | |
| MODIS normalized difference vegetation index (NDVI) mean for 2018 | n = 6,134 | | 0.90 | * | |
| 1st tertile [0.06; 0.3 [(low) | 22 (1.07) | Reference | | | |
| 2nd tertile [0.3; 0.4 [(medium) | 65 (3.13) | 1.1 (0.5–2.1) | 0.86 | | |
| 3rd tertile [0.4; 0.6) (high) | 39 (1.96) | 0.9 (0.4–2.0) | 0.87 | | |
| Aridity index | n = 6,134 | | 0.54 | * | |
| 1st tertile [0.59; 0.61 [(low) | 15 (0.72) | Reference | | | |
| 2nd tertile [0.61; 0.62 [(medium) | 92 (4.38) | 0.6 (0.2–1.6) | 0.33 | | |
| 3rd tertile [0.65; 0.65] (high) | 19 (0.97) | 1.0 (0.3–3.3) | 0.96 | | |

Notes

† Adjusted Generalized logistic mixed model estimating equations with exchangeable correlation structure.

* Variable dropped from fully adjusted model during model adjustment process using lowest AIC criteria.

** Variable in the final adjusted model but with no significant category

Abbreviation: School Aged Children (SAC), Pre School Aged Children (PSAC), confidence interval (CI), Moderate Resolution Imaging Spectroradiometer (MODIS)

antenatal care starting from the second trimester of pregnancy. [27] In this context, we sought to better understand patterns of STH infection in order to move towards the elimination of STH as a public health problem, by reaching a prevalence of STH less than 1%, as prescribed by the WHO NTD Roadmap and London Declaration on NTD. [10,26]

When focusing on the at-risk population of SAC, the prevalence of STH infection found in the current study is lower than that reported in the same district using the same diagnostic technique in 2015 during the national mapping exercise (5.2% versus 20.0% respectively, p<0.001). [13] That survey was conducted with a total of 250 stool samples from SAC collected from schools located in 5 rural villages. The prevalence of infections with *Ascaris lumbricoides* (3.0% *versus* 15.6% respectively, p<0.001) or *Trichuris trichiura* (0.15% in 2018 *versus* 4.8% in 2015, p<0.001) decreased while the decrease in prevalence of hookworm in SAC was less marked (2.0% in 2018 versus 4.0% in 2015, p = 0.054) compared to the findings of the national STH mapping three years earlier. [13] The decline in STH prevalence in the study area might be related to differences in sampling, as the current study was conducted in the community instead of schools, with more than 6,000 stools randomly selected from three age groups (PSAC, SAC and adults). [11] Soil-transmitted helminths prevalence estimates can vary depending on the sampling strategies used. [28] The reasons for the observed variations of prevalence between hookworm and other STH species in SAC could also be that the STHs' rate of reinfection post-treatment varies across species, with a faster reinfection with *Ascaris lumbricoides* than hookworm. A systematic review of helminth reinfection at 3, 6, and 12 months, after drug treatment shows that *Ascaris lumbricoides* prevalence reached 26%, 68% and 94% of pretreatment levels, respectively and for hookworm, 30%, 55% and 57%. [29] These results may also be partly explained by the fact that the current STH program does not include adults. The suggestion that hookworm prevalence only decreased slightly between 2015 and 2018, may be due to the persistent untreated adult reservoir in which hookworm are most common. Data from several worm expulsion studies show that the proportion of

hookworms harbored by adults ranges from 70 to 85%, [30–35] and a reinfection–infection study in Indonesia shows that adults have higher reinfection rates with hookworm than children. [36] Children cleared of hookworms through annual school de-worming could easily be re-infected at home through contact with adult members of their households.

Hookworm prevalence was higher in adults while *Ascaris lumbricoides* prevalence was higher in children. One explanation of these findings might be helminth species transmission modes. [37] The three species of STHs (*A. lumbricoides*, *T. trichiura*, hookworm spp.) have relatively similar cycles involving the presence of adult worms in the intestine. However the main mode of transmission of *Ascaris lumbricoides* and *Trichuris trichiura* is through contaminated food and water (parasite egg ingestion) whereas hookworm are mainly transmitted by skin penetration, although they can be transmitted by ingestion. [38,39] The eggs of *Ascaris lumbricoides* and *Trichuris trichiura* are found in soil contaminated by human feces or in uncooked food contaminated by soil containing eggs of the worm. A person becomes infected after accidentally swallowing the fertile eggs. Children may be more likely to be infected with *Ascaris lumbricoides* because they are more likely to put their contaminated fingers in their mouths after playing in contaminated soil. [40] Unlike *Ascaris lumbricoides* infection, which declines in prevalence with age, hookworm infects all ages throughout life with prevalence increasing in adults. [30]

Community-level prevalence and the arithmetic mean of infection intensity were significantly correlated for all STHs infections in our study, with a strong prevalence-intensity correlation for infection with hookworm and *Ascaris lumbricoides*. Similar trends were recently found in Kenya for hookworm and *Trichuris trichiura*. [24] At the individual level, prevalence and intensity of hookworm infection followed the same trend. Markers of poverty and exposure to environmental sources of STHs infections, including being a farmer, lack of improved or private sanitation facilities, low income, poor access to water, no or limited education, or living in a house with natural floor material were all associated with a higher prevalence or intensity of hookworm infection. These findings are linked with the mode of hookworm transmission, which is direct either by ingestion (for *A. duodenale*) or by skin penetration (both *N. americanus* and *A. duodenale*) of infective larval stages living in the soil. [41,42] These findings are consistent with the results of a recent study in Kenya where there was a strong association between hookworm infection prevalence and intensity and socio-economic status, with those in the poorest households having the heaviest infections and highest prevalence, and wealthier individuals having the lightest intensity and reduced odds of infection. [24] Globally, a negative correlation between hookworm infections and income level is demonstrated in cross-country comparisons. [43–45] Moderate population density, corresponding to a peri-urban environment, was also associated with both high prevalence and heavy intensity of hookworm infection when compared to the higher population density observed in urban environments. [46]

Although we found no association between hookworm prevalence and water source, the quality of water seems to affect the intensity of hookworm infections. Heavier intensity infections were found in participants with access only to unimproved water, such as unprotected wells, unprotected springs and surface water available at more than 30 minutes from the house. In a school survey in Togo, unimproved drinking water was associated with higher odds and intensity of hookworm. [47] Malaysian children with access to piped water were less infected with hookworm. [48] However, other researchers have found no statistically significant associations between piped water access and hookworm infection [49,50]. We did not find any association between WASH variables and either prevalence or intensity of *Ascaris lumbricoides* infection. However, there is evidence that integrated water, sanitation and hand hygiene intervention, treatment of water with chlorine [51], drinking piped water, as well as

hand washing before eating and after defecating reduce the odds of *Ascaris lumbricoides* infection. [52]

This study had a number of strengths, including the large population size, completeness and quality of data and the high level of quality control (QC) for Kato-Katz diagnosis. Actually Kato Katz's QC consisted in double reading by the lab technicians of the whole sample with an additional control of a subset of samples by a senior skilled parasitologist. However, this study does has some limitations. First, the STH prevalence in Come hides inter- and intra-specific variations between clusters that will be developed in further analyses once the parent study is unblinded and we have access to those results. It was also necessary to use a staged approach to sampling in order achieve the required number of participants who consented to participate in the longitudinal monitoring cohort, which may have limited its representativeness. Another possible limitation is the reliance on Kato-Katz to detect STH. Kato-Katz is poorly sensitive, particularly for low intensity infections [53,54] and can be affected by storage and processing time and methods [55]. In this regard, DeWorm3 field workers were all equipped with ice packs and cold boxes to shorten as much as possible the delay between stool sample production and preservation/refrigeration. Stool samples were examined by lab technicians within 30 minutes after receipt. Future analyses using qPCR-based methods will allow for more sensitive detection of STH in stool.

## Conclusion

This analysis of the DeWorm3 baseline study data shows that hookworm are the predominant STH in Comé, with a persistent reservoir in adults. This infection reservoir is not addressed by current school-based MDA control measures. These data suggest that community-based MDA may help eliminate STH as a public health problem. WASH should be improved because we found that improved unshared sanitation and access to improved water sources are associated with lower prevalence and/or intensity of hookworm infection. Programmatic efforts should pay particular attention to farmers and populations living in poverty in urban, rural and peri-urban environments. The DeWorm3 trial (2017–2022) will determine the feasibility of STH transmission interruption through community-wide MDA given twice-a-year for three years in this settings. With these results we will inform programmatic and policy decisions to improve efforts to eliminate morbidity and infection due to these pervasive infections.

## Supporting information

**S1 Strobe checklist. STROBE Statement—Checklist of items that should be included in reports of *cross-sectional studies*.**
(DOC)

**S1 Table. Summary of Kato-Katz Quality Assurance (QA).**
(DOCX)

**S2 Table. Burden of moderate-to-heavy intensity (MHI) STH infection in the study population by age group, during DeWorm3 baseline analysis in Comé, Bénin.**
(DOCX)

**S3 Table. Burden of moderate-to-high intensity STH infection among infected individuals by age group and gender, during DeWorm3 baseline analysis in Comé, Bénin.**
(DOCX)

**S4 Table. Factors univariately associated with hookworm infection in Comé, Bénin: findings from DeWorm3 cluster randomized trial baseline pre-treatment survey using**

**generalized logistic mixed model.**
(DOCX)

**S5 Table. Intra-Class Correlation values.**
(DOCX)

## Acknowledgments

Thanks to Kate Halliday for sharing a Stata code to illustrate the age-infection profile for hookworm and *Ascaris lumbricoides.* We are particularly grateful to William Oswald, David Kennedy and Mira Emmanuel-Fabula for their invaluable contributions to data collection and quality.

We thank Wilfrid Batcho, Coordinator of the National NTD Program (PNLMT) and his staff for their continuous support, and finally we are grateful to all Comé district population, administrative and health authorities for their support in the success of this study.

## Author Contributions

**Conceptualization:** Arianna Rubin Means, D. Timothy J. Littlewood, Judd L. Walson, Kristjana Hrönn Ásbjörnsdóttir.

**Data curation:** Euripide F. G. A. Avokpaho, Parfait Houngbégnon, Eloïc Atindégla, David S. Kennedy, Sean R. Galagan, Kristjana Hrönn Ásbjörnsdóttir.

**Formal analysis:** Euripide F. G. A. Avokpaho, Sean R. Galagan, Kristjana Hrönn Ásbjörnsdóttir.

**Funding acquisition:** D. Timothy J. Littlewood, Judd L. Walson.

**Investigation:** Euripide F. G. A. Avokpaho, Parfait Houngbégnon, Manfred Accrombessi, Eloïc Atindégla, Gilles Cottrell, Moudachirou Ibikounlé, Adrian J. F. Luty.

**Methodology:** Arianna Rubin Means, David S. Kennedy, D. Timothy J. Littlewood, Sean R. Galagan, Judd L. Walson, Kristjana Hrönn Ásbjörnsdóttir.

**Project administration:** Elodie Yard, Achille Massougbodji, Judd L. Walson, Adrian J. F. Luty.

**Resources:** Moudachirou Ibikounlé, Adrian J. F. Luty.

**Software:** David S. Kennedy.

**Supervision:** Parfait Houngbégnon, Manfred Accrombessi, Elodie Yard, Arianna Rubin Means, André Garcia, Achille Massougbodji, Judd L. Walson, Moudachirou Ibikounlé, Kristjana Hrönn Ásbjörnsdóttir, Adrian J. F. Luty.

**Validation:** André Garcia, Sean R. Galagan, Judd L. Walson, Moudachirou Ibikounlé, Kristjana Hrönn Ásbjörnsdóttir, Adrian J. F. Luty.

**Visualization:** David S. Kennedy.

**Writing – original draft:** Euripide F. G. A. Avokpaho.

**Writing – review & editing:** Parfait Houngbégnon, Manfred Accrombessi, Eloïc Atindégla, Arianna Rubin Means, D. Timothy J. Littlewood, André Garcia, Achille Massougbodji, Sean R. Galagan, Judd L. Walson, Gilles Cottrell, Moudachirou Ibikounlé, Kristjana Hrönn Ásbjörnsdóttir, Adrian J. F. Luty.

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
