## [Decision Letter · Decision Letter 0]

29 Jan 2021

Dear Dr Avokpaho,

Thank you very much for submitting your manuscript "Factors associated with soil-transmitted helminth infection in Benin: findings from the DeWorm3 study" for consideration at PLOS Neglected Tropical Diseases. As with all papers reviewed by the journal, your manuscript was reviewed by members of the editorial board and by several independent reviewers. The reviewers appreciated the attention to an important topic. Based on the reviews, we are likely to accept this manuscript for publication, providing that you modify the manuscript according to the review recommendations. 

Three independent reviewers recommended minor revisions to your manuscript and I agree. Please address the points raised by each reviewer by point-by-point basis and resubmit a revised manuscript.

Sincerely,

jong-Yil Chai

Associate Editor

Marco Coral-Almeida

Deputy Editor

Three independent reviewers recommended minor revisions to your manuscript and I agree. Please address the points raised by each reviewer by point-by-point basis and resubmit a revised manuscript.

Reviewer's Responses to Questions

**Key Review Criteria Required for Acceptance?**

**Methods**

-Are the objectives of the study clearly articulated with a clear testable hypothesis stated?

-Is the study design appropriate to address the stated objectives?

-Is the population clearly described and appropriate for the hypothesis being tested?

-Is the sample size sufficient to ensure adequate power to address the hypothesis being tested?

-Were correct statistical analysis used to support conclusions?

-Are there concerns about ethical or regulatory requirements being met?

Reviewer #1: The methods are mostly very clear. Please see my comments and suggestions in the pdf.

Reviewer #2: - Please provide information about the data sources as supplementary description (not use reference).

Reviewer #3: The methodology of the study is quite appropriate. The following concerns need to be addressed:

1. Why was qPCR not used to assess the stool parasites as PCR has been shown to be more sensitive than microscopy?

2. How was the sample size calculated for this study?

3. How was the sampling done - random sampling or any other sampling method?

**Results**

-Does the analysis presented match the analysis plan?

-Are the results clearly and completely presented?

-Are the figures (Tables, Images) of sufficient quality for clarity?

Reviewer #1: I believe the analysis does match the analysis oplan. The tablets need considerable improvement for clarity. Please see my comments and suggestions in the pdf.

Reviewer #2: - Please provide the detailed map of research area in this paper not reference.

- Transfer the contents of Descriptive in the Results part to the study design in Materials and Methods part. 

- The interesting point in the table 4 is that Ascaris lumbricoides was more prevalent in fisher (not farmer). Please describe your opinion in the discussion part.

Reviewer #3: The results as described are fine. The study does not examine the association of any hematological parameters with stool parasites including anemia. This is a major drawback.

**Conclusions**

-Are the conclusions supported by the data presented?

-Are the limitations of analysis clearly described?

-Do the authors discuss how these data can be helpful to advance our understanding of the topic under study?

-Is public health relevance addressed?

Reviewer #1: The conclusions are supported by the data and the limitations are well described. The authors do discuss (although mostly in the conclusions) how their findings could help improve the control of STH. Please see my comments and suggestions in the pdf.

Reviewer #2: (No Response)

Reviewer #3: The conclusions are appropriate.

**Editorial and Data Presentation Modifications?**

Reviewer #1: Please see my comments and suggestions in the pdf.

Reviewer #2: - Use the name of any parasites according to the normal journal style (Italic, abbreviation etc.) or uniform style determined by the author in whole manuscript including tables, supplementary data, etc.

Reviewer #3: None.

**Summary and General Comments**

Reviewer #1: This study confirms several of the risk factors assumed to be related to STH infections. It was very nice to see that so many logical associations were found. The manuscript is, overall, well written but I submit (pdf) some suggestions on how to improve it.

Reviewer #2: (No Response)

Reviewer #3: The study examines the presence of STH and associated socio-economic, environmental and WASH factors. The concerns include lack of any hematological details, lack of examining other risk factors etc.

PLOS authors have the option to publish the peer review history of their article (what does this mean?). If published, this will include your full peer review and any attached files.

Reviewer #1: No

Reviewer #2: No

Reviewer #3: No
---

## [Decision Letter · Decision Letter 1]

1 Jun 2021

Dear Dr Avokpaho,

Thank you very much for submitting your manuscript "Factors associated with soil-transmitted helminths infections in Benin: findings from the DeWorm3 study" for consideration at PLOS Neglected Tropical Diseases. As with all papers reviewed by the journal, your manuscript was reviewed by members of the editorial board and by several independent reviewers. The reviewers appreciated the attention to an important topic. Based on the reviews, we are likely to accept this manuscript for publication, providing that you modify the manuscript according to the review recommendations. 

One reviewer suggested some minor revisions to your revised manuscript. Please see the comments and address the points and resubmit. Thanks for your kind cooperation.

Sincerely,

jong-Yil Chai

Associate Editor

Marco Coral-Almeida

Deputy Editor

One reviewer suggested some minor revisions to your revised manuscript. Please see the comments and address the points and resubmit. Thanks for your kind cooperation.

Reviewer's Responses to Questions

**Key Review Criteria Required for Acceptance?**

**Methods**

-Are the objectives of the study clearly articulated with a clear testable hypothesis stated?

-Is the study design appropriate to address the stated objectives?

-Is the population clearly described and appropriate for the hypothesis being tested?

-Is the sample size sufficient to ensure adequate power to address the hypothesis being tested?

-Were correct statistical analysis used to support conclusions?

-Are there concerns about ethical or regulatory requirements being met?

Reviewer #2: (No Response)

Reviewer #3: (No Response)

Reviewer #4: -The objectives of the study are clearly articulated 

-Since the study is part of a longitudinal study and that the current report was only on the baseline data, the design appropriate should be focus on a cross sectional design, irrespective of the main study. So this should be specified or clarified 

-The study population need more details and not just a reference to a published manuscript. Particularly activities carried out by the population which might be a risk factor for STH infection, as well as the Socio-economic level of the study population.

-The sample size is robust enough .

-The author chose adult group >15 years old, can they give a rational underline this choice? 

Indeed In the same document 14-18 years old are considered as adolescent,. This can confuse a reader, if rational was not provided.

-Only Kato-katz technique was performed in this study, while for hookworm, more sensitive stool culture tests are available and recommended including McMaster technique, or Harada-Mori which provide more comprehensive results. 

We can understand that in a routine RT-PCR may not be used, but umblinding reason provided by the author should be applied to a microscopy as well. Please provide more information on the study limitations regarding the use of the kato-katz only in a discussion section.

**Results**

-Does the analysis presented match the analysis plan?

-Are the results clearly and completely presented?

-Are the figures (Tables, Images) of sufficient quality for clarity?

Reviewer #2: (No Response)

Reviewer #3: (No Response)

Reviewer #4: The analysis presented match well the analysis plan. However, I have a concern regarding the age groups. Are there any rational for choosing these different age groups?

The parenthesis in the tables should indicating the inclusion or exclusion of the numbers with parenthesis. 

Why authors can't create more groups in adult group with aim to yield the ttrend association of hookworm infection with age if any? 

On Fig.4 it is not clear that hookworm infection increases with age. 

Is there any explanation related to the prevalence difference between male and female?

For Ascaris, here again there is a cluster increase of Ascaris around 10 years old, indicating no clear negative association between Ascaris infection with age. This is true only after 15 years old

Which Hookworm species authors reported here?

**Conclusions**

-Are the conclusions supported by the data presented?

-Are the limitations of analysis clearly described?

-Do the authors discuss how these data can be helpful to advance our understanding of the topic under study?

-Is public health relevance addressed?

Reviewer #2: (No Response)

Reviewer #3: (No Response)

Reviewer #4: The conclusion supported the data presented partially. Indeed only hookworm infection was retained the attention of the autors.

For limitation, author should elaborate on the limitation of Kato-katz as well as on available microscopic techniques including stool cultures which may change the current result particularly when hookworm is concern.?

**Editorial and Data Presentation Modifications?**

Reviewer #2: (No Response)

Reviewer #3: (No Response)

Reviewer #4: Minor revision are needed,.the choice of age groups, the limitation of Kato-katz, and other stool microscopical testing.

**Summary and General Comments**

Reviewer #2: (No Response)

Reviewer #3: (No Response)

Reviewer #4: This manuscript is well written and rThe author reports the already known factor associated with STH infection, but has an advantage with a high sample size. Minor chnages are required and can then been published in the PNTD. Indeed it provides a lot of update on the major STH infections.

PLOS authors have the option to publish the peer review history of their article (what does this mean?). If published, this will include your full peer review and any attached files.

Reviewer #2: No

Reviewer #3: No

Reviewer #4: No

Figure Files:

Data Requirements:

Reproducibility:

References

---

## [Editor Report · Decision Letter 2]

13 Jul 2021

Dear Dr Avokpaho,

We are pleased to inform you that your manuscript 'Factors associated with soil-transmitted helminths infections in Benin: findings from the DeWorm3 study' has been provisionally accepted for publication in PLOS Neglected Tropical Diseases.

Best regards,

jong-Yil Chai

Associate Editor

Marco Coral-Almeida

Deputy Editor

Your revised manuscript has been reviewed and the reviewers gave no further comments. I am happy to decide to accept your revised manuscript.

---

## [Editor Report · Acceptance letter]

12 Aug 2021

Dear Dr Avokpaho,

We are delighted to inform you that your manuscript, "Factors associated with soil-transmitted helminths infection in Benin: findings from the DeWorm3 study," has been formally accepted for publication in PLOS Neglected Tropical Diseases.

Best regards,

Shaden Kamhawi

co-Editor-in-Chief

Paul Brindley

co-Editor-in-Chief
